



# The impact of hydrothermal vent geochemistry on the addition of iron to the deep ocean

Alastair J. M. Lough[1,*], Alessandro Tagliabue[2], Clément Demasy[1], Joseph A. Resing[3], Travis Mellett[4], Neil J. Wyatt[1], Maeve C. Lohan[1]

[1]Ocean & Earth Science, University of Southampton, Southampton, SO14 3ZH, UK
[2]Earth, Ocean & Ecological Sciences, Liverpool, L69 3BX, UK
[3]Cooperative Institute for Climate, Oceans, and Ecosystem Studies, University of Washington and NOAA-PMEL, Seattle, USA
[4]University of South Florida, St. Petersburg, FL, FL 33701, United States.

*Correspondence to*: Alastair J.M. Lough (Now at the University of Leeds: A.J.M.Lough@leeds.ac.uk)

**Abstract.** Supply of iron (Fe) to the surface ocean supports primary productivity and while hydrothermal input of Fe to the deep ocean is known to be extensive, it remains poorly constrained. Global estimates of hydrothermal Fe supply rely on using the dissolved Fe (dFe) to excess He (xs$^3$He) ratios to upscale fluxes, but observational constraints on dFe/xs$^3$He may be sensitive to assumptions linked to sampling and interpolation. We examined the variability in dFe/xs$^3$He using two

methods of estimation, for four vent sites with different geochemistry along the Mid-Atlantic Ridge. At both Rainbow and TAG, the plume was sampled repeatedly and the range of dFe/xs$^3$He was 4 to 63 and 4 to 87 nmol/fmol, respectively, primarily due to differences in plume age. To account for background xs$^3$He and shifting plume position, we calibrated He values using contemporaneous dissolved Mn (dMn). Applying this approach more widely, we found dFe/xs$^3$He ratios of 12, 4-8, 4-44, 4-86 nmol/fmol for the Menez Gwen, Lucky Strike, Rainbow and TAG hydrothermal vent sites, respectively.

Differences in plume dFe/xs$^3$He across sites were not simply related to the vent end member Fe and He fluxes. Within 40 km of the vents, the dFe/xs$^3$He ratios decreased to 3-38 nmol/fmol, due to the precipitation and subsequent settling of particulates. The ratio of colloidal Fe to dFe was consistently higher (0.67-0.97) than the deep N. Atlantic (0.5) throughout both the TAG and Rainbow plumes, indicative of Fe exchange between dissolved and particulate phases. Our comparison of TAG and Rainbow shows there is a limit to the amount of hydrothermal Fe released from vents that can form colloids in the

rising plume. Higher particle loading will enhance the longevity of the Rainbow hydrothermal plume within the deep ocean assuming particles undergo continual dissolution/disaggregation. Future studies examining the length of plume pathways required to escape the ridge valley will be important in determining Fe supply from slow spreading mid-ocean ridges to the deep ocean, along with the frequency of ultramafic sites such as Rainbow. Resolving the ridge valley bathymetry and accounting for variability in vent sources in global biogeochemical models will be key to further constraining the

hydrothermal Fe flux.



# 1 Introduction

Iron (Fe) is an essential trace element that shapes ocean biogeochemical cycles. Photosynthetic primary productivity and nitrogen fixation in the surface ocean depend on the supply of Fe from lithogenic sources. Predicting the extent to which primary productivity is dependent on Fe supply is limited by our understanding of Fe sources and sinks in the open ocean

(Tagliabue et al., 2017). This is particularly important in Fe limited regions such as the Southern Ocean, where changes in the supply of Fe to the surface ocean may dramatically shift the Earth's atmospheric $CO_2$ content (Gottschalk et al., 2019) and where hydrothermal vents may play an important role as a source of Fe (Tagliabue, 2010; Tagliabue and Resing, 2016; Ardyna et al., 2019; Weber, 2020; Schine et al., 2021).

The magnitude and importance of Fe supplied from different sources (i.e. glaciers, rivers, aerosols, sediments and

hydrothermal vents) is an ongoing subject of debate. In the last 15 years, the role of hydrothermal vents in supplying Fe to the deep ocean, that may subsequently upwell in the Southern Ocean, has received significant attention, with questions surrounding the biogeochemical processes that could facilitate long range transport of Fe from the seafloor (Toner et al., 2009; Tagliabue, 2010; Yucel et al., 2011; Saito et al., 2013; Resing et al., 2015; Fitzsimmons et al., 2017). In order to examine the hydrothermal flux of Fe to the deep ocean, changes in Fe concentration are frequently compared to excess

Helium ($xs^3He$), derived from $\delta^3He$, which is an inert tracer of hydrothermal activity (Lupton and Craig, 1981; Wu et al., 2011; Saito et al., 2013; Resing et al., 2015; Fitzsimmons et al., 2017). Primordial Helium (He) degasses from the Earth's mantle and as a result hydrothermal fluids are enriched in $^3He$ relative to background seawater (Lupton et al., 1977). As He is an unreactive dissolved gas it is an ideal source tracer for hydrothermal plumes. The ratio of Fe to $xs^3He$ has been used as a basis for modelling the impact of hydrothermal Fe on the ocean iron cycle and surface ocean primary productivity plus the

associated carbon export (Tagliabue, 2010).

Recent field studies have found a linear relationship between dissolved Fe (dFe) and $xs^3He$, interpreted as conservative behaviour of Fe. In some cases, Fe appears to behave conservatively over thousands of kilometres, while in others the conservative relationship of $dFe/xs^3He$ is only apparent over the ridge (Saito et al., 2013; Resing et al., 2015). The observation of conservative behaviour was unexpected for a reactive metal such as Fe, as previous studies working at the <1

km scale had estimated that up to ~90 % of Fe released from seafloor vents precipitates as Fe-sulphide and Fe-oxyhydroxide mineral particles, as the Fe and hydrogen sulphide ($H_2S$) rich vent fluids are released into cold, well-oxygenated, deep ocean waters (German et al., 1991; Field and Sherrell, 2000; Severmann et al., 2004). It is the remaining hydrothermal Fe that does not form fast settling mineral particles that is ultimately exported, as an effective flux to the deep ocean of fine colloidal particles and/or organic Fe complexes (Bennett et al., 2008; Hawkes et al., 2013; Kleint et al., 2016). It is thought that the

off-axis linear relationship of dFe with $xs^3He$ arises because dFe species formed in the plume exhibit relatively unreactive behaviour (Bennett et al., 2008; Yucel et al., 2011). An alternative hypothesis is that Fe is added to the dissolved fraction continuously by the dissolution/disaggregation of larger particulate phases as the plume disperses (Fitzsimmons et al., 2017), at a rate that maintains the $dFe/xs^3He$ ratio giving the appearance of conservative behaviour.





Studies that have used xs$^3$He as a tool for understanding hydrothermal Fe have typically sampled at the basin scale whereas
studies focusing on the <1 km scale tend to use other shorter lived tracers such as dissolved manganese (dMn) (James and
Elderfield, 1996; Field and Sherrell, 2000; Lough et al., 2017; Lough et al., 2019b; Lough et al., 2019a), magnesium
(Findlay et al., 2015) or rare earth element anomalies (Severmann et al., 2004). Furthermore, the Fe and He sampled at the
basin scale may represent an amalgamation of several vent sources from a ridge or several ridge crests, whereas the studies at
<1 km scale focus on Fe released from individual or at least fewer vent sites. Different vent sites are known to display
substantial variations in dFe/xs$^3$He ratios (Table 1)(Tagliabue et al., 2010) but the extent to which sampling scale, strategy
and use of different tracers affects the interpretation of the effective hydrothermal iron flux is a barrier to further refining the
conceptual and numerical models we rely on for larger scale integration.

To address this knowledge gap, this study sampled hydrothermal plumes along the same ridge from multiple vent sources at
a scale of 10's of km's, using both short lived (dMn, weeks (Cowen et al., 1990; Trocine and Trefry, 1988; Field and
Sherrell, 2000; Massoth et al., 1994; Lavelle et al., 1992)) and long lived (xs$^3$He) as conservative tracers. We examined the
variability in dFe/xs$^3$He produced from different methods of estimation (Saito et al., 2013; Resing et al., 2015; Fitzsimmons
et al., 2017) in plumes originating from four vent sites along the Northern part of the Mid-Atlantic Ridge (MAR)
(GEOTRACES GA13 section). These vents cover a range of geological settings, plume dFe concentrations and importantly
Fe/H$_2$S ratios, which have been shown to correlate with colloid concentration in nascent plumes (i.e. 1-2 m above the vents)
(Gartman et al., 2014). Calculated Fe/xs$^3$He values are used to compare the separation of Fe between particulate-dissolved
fractions for the TAG and Rainbow plumes as they disperse within the ridge valley.

**Table 1. Summary of average endmember vent fluid data taken from the InteRidge database (Beaulieu et al., 2013a)**

| Vent field | Geology | T (ºC) | Cl⁻ (mM) | pH | He³ fmol kg⁻¹ | Fe nmol kg⁻¹ | Mn nmol kg⁻¹ | H$_2$S (mM) | dFe/xsHe (nmol/fmol) | dFe/H$_2$S |
|---|---|---|---|---|---|---|---|---|---|---|
| Menez Gwen | E-MORB | 285 | 335 | 4.4 | $2.0 \times 10^4$ | $1.4 \times 10^4$ | $7.1 \times 10^4$ | 1.53 | 1 | 0.01 |
| Lucky Strike | E-MORB | 301 | 493 | 3.5 | $1.0 \times 10^4$ | $5.61 \times 10^5$ | $2.62 \times 10^5$ | 3.08 | 47 | 0.18 |
| Rainbow | Serpentinite, gabbro, MORB | 366 | 750 | 2.9 | $2.5 \times 10^4$ | $2.41 \times 10^7$ | $1.96 \times 10^6$ | 0.93 | 962 | 25.86 |
| TAG | MORB | 359 | 661 | 3.2 | $1.8 \times 10^4$ | $5.11 \times 10^6$ | $5.52 \times 10^5$ | 5.13 | 249 | 1.00 |

**MORB = Mid-Ocean Ridge Basalt, E-MORB = Enriched Mid-Ocean Ridge Basalt.**

**Endmember vent fluid data from published studies is calculated by extrapolating to 0 Mg concentration (Douville et al., 2002).**






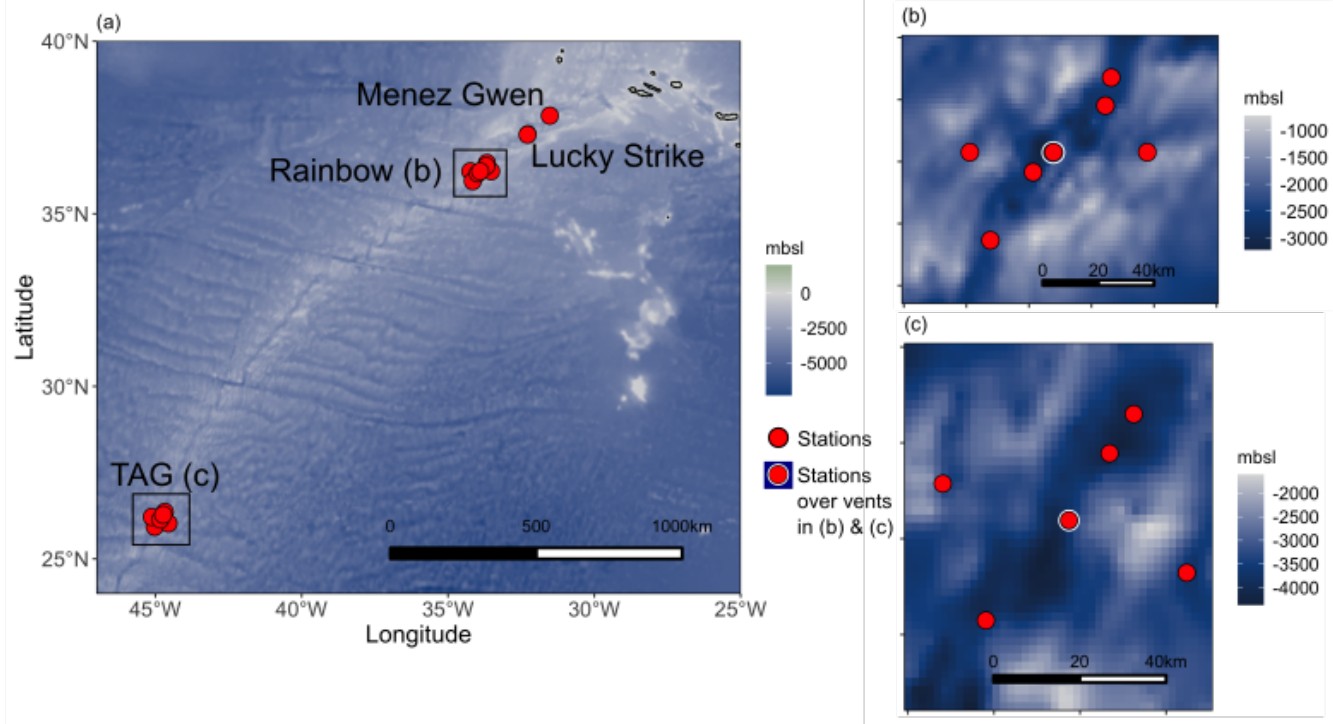

**Figure 1: Map showing the location of the main vent sites sampled during the GA13 voyage (a) and the nearby stations within 40 km of each site at (b) Rainbow and (c) TAG. Bathymetry data is sourced from The National Oceanic and Atmospheric Administration (NOAA) ETOP1. Maps were produced in R studio using the marmap package.**

## 2. Methods

### 2.1 Sample collection

The UK GEOTRACES GA13 voyage sailed along the northern Mid-Atlantic Ridge (MAR) during its passage from Southampton, UK, to Guadeloupe, France. The fieldwork campaign sampled the rising buoyant and neutrally-buoyant hydrothermal plumes of a set of known hydrothermal vent sites along the northern MAR (Fig. 1). At each site, hydrothermal plumes were detected using a combination of sensors. A traditional CTD package (*Seabird 911*) was used to identify anomalous changes in salinity and temperature relative to background N. Atlantic waters. Bespoke light scattering (LSS) and oxidation-reduction potential (ORP) sensors were used to identify particle dense plumes and young plume waters containing reducing chemical species (i.e. $Fe^{2+}$, $HS^-$ and $H_2$). Sampling casts were repeated over the TAG and Rainbow sites to examine the reproducibility of plume sampling relative to tidal forces and bottom currents that shift the plumes position in the water 100 column.

As part of the GEOTRACES programme, Fe was sampled according to the detailed procedures described previously (Cutter et al., 2010) and that we briefly outline below. However, He and Mn samples were collected from a standard (stainless steel)





rosette, and we show that Mn can be sampled cleanly without a clean lab or a titanium rosette frame (Supplementary Information; Fig. S1).

Helium sampling methods followed those described in Jenkins et al. (2015b). Briefly, 30 inches of copper pipe was rinsed several times through with seawater collected from Niskin bottles (*Ocean Test Equipment*) on the standard rosette using plastic tubing (*Tygon*). Once all air bubbles had been removed, the copper tubing was crimped at both ends to seal the pipe and sent for analysis at Woods Hole Oceanographic Institution.

$$\delta^3 He = \left( \frac{\left( ^3He/\ ^4He \right)_X}{\left( ^3He/\ ^4He \right)_{Air}} - 1 \right) \times 100\ \% \tag{1}$$

$$xs^3He = \left( \delta\ ^3He - -1.7 \right) \div 100 \times [He] \times 1.384 \times 10^{-6} \tag{2}$$

The $\delta^3$He isotope anomaly is defined relative to an atmospheric standard in Eq. (1) with $\delta^3$He expressed in percent. The $xs^3$He represents the mantle derived $^3$He that is approximate to the non-atmospheric $^3$He over saturation. The He isotope ratio anomaly 1.384 x $10^{-6}$ is the atmospheric $^3$He/$^4$He ratio, [He] is the molar concentration of He and –1.7 is the solubility
equilibrium constant (Jenkins et al., 2015a).

Seawater samples for trace metal analysis were collected using a titanium-frame CTD with 24 trace metal clean, 10 L, Teflon-coated Niskin bottles (*Ocean Test Equipment*) deployed on a plasma rope. A Sartobran 300 (*Sartorius*) filter capsule (0.2 µm) was used to filter seawater into clean low-density polyethylene (LDPE) bottles for dissolved trace metals. A separate aliquot of 0.2 µm filtered seawater was further filtered through 0.02 µm syringe filters (Anotop, *Whatman*) into
LDPE bottles for soluble Fe (sFe) (Ussher et al., 2010). Unfiltered seawater samples were collected for total dissolvable (TD) metals. All samples were acidified onboard to 0.024 M (UpA HCl, *Romil*).

### 2.2 Sample analysis

Dissolved and total dissolvable samples were analysed on-shore for Fe and Mn by ICP-MS (*Thermoscientific, Element XR*) using a standard addition method (Lough et al., 2017). Certified values for GEOTRACES reference material D2 (0.96 nM Fe
and 0.36 nM Mn) compared well with our measured values of $0.95 \pm 0.06$ nM Fe and $0.34 \pm 0.03$ nM Mn ($n = 6$). ICP-MS analysis of GSC reference material (measured GSC: $2.04 \pm 0.03$ nM Mn and $1.48 \pm 0.13$ nM Fe $n = 3$) also compared well with the preliminary consensus values (GSC $2.18 \pm 0.08$ nM Mn, $1.54 \pm 0.12$). In house standards with higher concentrations of Fe and Mn in the range of hydrothermal samples were measured repeatedly with relative standard deviations of 6 % for Mn and 7 % for Fe. Soluble Fe was measured by flow injection analysis with chemiluminescence detection (Obata et al.,
1993; Kunde et al., 2019a) with measured values of $0.94 \pm 0.04$ ($n = 6$) for D2 reference material. Measurements of GSP and GSC reference materials using flow injection also agree with the preliminary consensus values (consensus: GSP $0.16 \pm 0.05$, GSC $1.54 \pm 0.12$ nM, measured GSP $0.15 \pm 0.01$ nM $n = 7$, GSC $1.52 \pm 0.06$, $n = 10$). Colloidal Fe (cFe) is operationally defined as the difference between dFe (<0.2 µm) and sFe (<0.02 µm). Apparent particulate Fe (appPFe) is further operationally defined as the difference between TDFe (unfiltered) and dFe (<0.2 µm).



Dissolved Mn samples from the standard rosette were analysed at sea by flow injection analysis with in-line pre-concentration on resin-immobilized 8-hydroxyquinoline and colorimetric detection (Resing and Mottl, 1992). The SAFe reference samples were analysed simultaneously during sample analysis to determine the accuracy and precision of the method giving results for SAFe S, $0.82 \pm 0.06$ nM (n = 19; consensus value = $0.79 \pm 0.06$ nM); for SAFe D2, $0.41 \pm 0.03$ nM (n = 18; consensus value = $0.35 \pm 0.05$).

## 3. Results & Discussion

### 3.1 Quantifying Fe/$^3$He ratios

A drawback to using xs$^3$He as a tracer of hydrothermal Fe is that Fe, under GEOTRACES protocols, is sampled using trace-metal clean bottles mounted on a trace metal clean rosette, while He is sampled separately from a standard rosette to avoid metal contamination from Cu tubes used to collect $^3$He. At the Fe concentrations observed close to the vent sites, such

caution is likely unwarranted, however, to trace the full reach of a hydrothermal plume, trace concentrations 0.1 nM above background concentrations need to be detected. The best way to guarantee this resolution is to follow GEOTRACES trace-metal clean sampling protocols, as a result, Fe and $^3$He are never sampled from the same sampling bottle, same cast or at the same time (Fig. 2). Furthermore, sampling $^3$He requires the Cu pipes to be flushed with copious amounts of sample water, which would leave limited water available from the trace metal clean rosette to sample for trace metal concentrations, their

isotopes and chemical speciation (samples collected during GA13 that will be discussed in future publications). Given the complex physical dynamics of a dispersing plume within a ridge valley (Vic et al., 2018; Lahaye et al., 2019), sampling the same point in the plume twice is near impossible. Therefore, here we apply three different ways of calculating dFe/xs$^3$He to assess which best represent the plume:

1. Plume integration method: Integrate dFe and xs$^3$He data across the plume from samples taken from each cast (e.g trace-

metal clean and standard rosette) (Resing et al., 2015; Fitzsimmons et al., 2017). This approach assumes that multiple depths through the plume have been sampled on both casts that are representative of a vertical cross section of the plume. In a sampling scenario such as that shown in Fig. 2 this approach is likely to lead to unrealistic dFe/xs$^3$He ratio's and is more suited to an off axis setting where the plume position is less variable.

2. Dual-Mn method: Constrain the xs$^3$He corresponding to the dFe data using measurements of dMn on both rosette systems.

This approach relies on the conservative behaviour of dMn over timescales of weeks (Cowen et al., 1986; Lough et al., 2017; Lough et al., 2019a) (Fig. S2) and uses the linear relationship between dMn/xs$^3$He measured from the standard rosette (Fig. 4A) to extrapolate the expected xs$^3$He for samples taken with the trace metal clean rosette. The dMn derived xs$^3$He values can then be integrated across the same samples as for dFe, which would account for between cast variability in the plume dynamics in a consistent manner. Furthermore, this site-specific approach helps us to account for any variability in

background xs$^3$He present in North Atlantic water masses, where decay of tritium from historic nuclear bomb tests has added $^3$He (Jenkins et al., 2015b).





**Figure 2: Schematic diagram illustrating the difficulties in sampling different elements at the same stage of a plume over a vent site using separate trace metal clean (a) and standard casts (b). The targets on the black lines represent depths sampled in the water column. The grey shaded area indicates where there are samples taken at the same depth for xs³He and dFe. Notice that at the same depth for different casts dFe is increasing with depth whilst xs³He is decreasing with depth due to the offset in the plume anomaly.**

3. A third method of estimation was explored using the depth profile of xs$^3$He on the standard rosette and interpolating between depths, to calculate xs$^3$He at the depths sampled by the trace metal clean rosette. This is similar to the approach used by Saito et al. (2013), however this gave significantly different results from the other two methods, generating negative numbers in some instances (Table S1). This is because it relies on the assumption that the xs$^3$He depth profile is the same on both sampling casts. While the assumption that the shape of the depth concentration profiles is unchanging between casts is



likely safe in an off-axis setting, Fig. 3 and Fig. S4 show that this cannot be assumed within the ridge valley. Thusly we
focus on the integration methods explained above.

Any samples with xs$^3$He <0.1 fM, dFe <0.5 nM, dMn <0.15 nM and neutral density $\leq$ 27 kg/m$^{-3}$ were excluded from analysis
as these waters are deemed to have not been influenced by hydrothermal activity. Profiles shown in Fig. 3 and the
supplement only show samples included in this analysis. The full data set can be viewed or downloaded through the
GEOTRACES international data product (Schlitzer et al., 2018)

Directly over the TAG and Rainbow sites, where the plume was sampled repeatedly, the range of dFe/xs$^3$He across the
integration methods was extensive, ranging from 4 to 87 and 4 to 63 nmol/fmol, respectively (Table 2). These values were
different even when the two methods are applied to data from the same casts. This is likely to be due to the two sampling
rosettes intersecting different areas of the plume (i.e margins or core) during casts and/or changes in plume depth over time
(time between standard and trace metal clean casts was 2-9 hours), despite the ship maintaining a steady surface ocean
position (Fig. 2 and 3). This degree of dFe/xs$^3$He variability between methods was also observed in the single station
estimates from the Menez Gwen and Lucky Strike locations, where ratios were 5 to 12 and 4 to 26 nmol/fmol, respectively
(Table 2), highlighting the different values that can be produced just by using a different method of calculating dFe/xs$^3$He.
The difference in calculated dFe/xs$^3$He ratios was consistently lower (maximum difference of 7 nmol/fmol) at stations away
from the main vent sites (Table S1). Hence, the variability in calculated dFe/xsHe directly over the vent sites is largely down
to the changing position of the plume over the vent site relative to the sampling rosette between casts, despite the ship
maintaining the same position (Fig. 2).

We focus on the dual-Mn method as the most robust means to estimate the dFe/xs$^3$He ratio. As it can account for differences
in position of the plume between sampling devices (Fig. 2 and 3) and background xs$^3$He. Applying site specific dMn/xs$^3$He
relationships from the standard rosette system to the dMn of the TMR rosette, the dual-Mn method finds dFe/xs$^3$He ratios of
12, 4-8, 4-44, 4-86 nmole/fmole at Menez Gwen, Lucky Strike, Rainbow and TAG respectively for the hydrothermal plumes
directly over these vent sites, (Table 2., Fig. 4C). Although the Dual-Mn method is less sensitive to differences across
sampling systems, large differences in dFe/xs$^3$He remain over the vent sites, with dFe/xs$^3$He still differing by a factor of 2-21
when comparing different casts. This serves to highlight the practical challenges of determining site specific Fe/xs$^3$He ratios,
even in a focussed study effort, as a result of complex bottom currents frequently shifting plume waters (Lahaye et al., 2019).







**Figure 3.** Depth profiles of Fe fractions over Rainbow (note the log scale on x-axis) from two separate trace metal clean (TMR) casts (a,b), dMn and xs$^3$He from two separate standard rosette (SSR) casts (b,e), and light scattering (LSS) and oxidation reduction potential (ORP) sensor profiles for the casts shown in a and b (c and f). There was no LSS sensor on the standard rosette for casts over Rainbow. The same profiles are shown (d, e, f) 34 km North of Rainbow.



**Table 2. Summary of plume dFe/xs³He ratios over vent sites and for repeat sampling at TAG and Rainbow using different methods to calculate dFe/xs³He.**

| Vent site (Station number) | Plume integrated dFe/xs³He (nmol/fmol) values | | | |
|---|---|---|---|---|
| | Integrated from separate casts (Method 1) | SR:TMR Sample depths integrated (n) (Method 1) | Dual-Mn method (Method 2) | TMR Sample depths integrated (n) (Method 2) |
| Menez Gwen (6) | 5 | 4:3 | 12 | 5 |
| Lucky Strike (7) | 26 | 5:4 | 4 | 7 |
| Lucky Strike (8) | ND | - | 8 | 12 |
| Rainbow (16) | 49 | 6:9 | 12 | 9 |
| Rainbow (38) | 35 | 6:3 | 44 | 3 |
| Rainbow* (38) | 63 | 3:2 | 4 | 2 |
| TAG (34) | 4 | 16:11 | 7 | 12 |
| TAG (35) | 87 | 17:11 | 86 | 15 |
| TAG (37) | ND | - | 4 | 15 |

**ND = no data, either no dMn data available from the trace metal rosette or no ³He data available from the standard rosette at the equivalent depth. The concentration depth profiles used for the integration at each station are shown in the supplementary Figures.**

**\*The young rising plume was identified over Rainbow close to the seafloor with density lower than that of other stations at the same depth (Supplementary Information, Fig. S8 and S10). This signal is separated as these samples will be from a cross section of**
**the young rising plume as the CTD rosette passed through it. An extended version of this table with data from all stations is presented in the supplementary information (Table S1).**

### 3.2 Linking water column Fe/xs³He ratios to vent fluid endmembers

The extent to which sample depths over the vent sites are representative of the core of the hydrothermal plume can be appraised by 1) assessing LSS and ORP sensor profiles, 2) comparing plume dMn and xs³He from the standard rosette, and
3) comparing TDFe to xs³He from the trace metal clean rosette (estimated from the dual-Mn method) with the respective vent fluid endmember ratios (Fig. 4). The vent fluid endmember dFe represents the majority of Fe released from the vent source because vent fluids have a low pH <2 and significant quantities of particulates are yet to form. In the neutrally buoyant plume emplaced directly over each vent site, limited particle settling will have occurred removing ~0-30 % Fe and ~0 % of the Mn present in the vent fluids (Mottl and Mcconachy, 1990; Severmann et al., 2004; Findlay et al., 2015; Lough
et al., 2019a).





**Figure 4.** Linear relationships between plume dMn and xs$^3$He at sites within 40 km of each vent source from the standard rosette (a). The data from (a) is depth integrated to compare the plume dMn/xs$^3$He with vent dMn/xs$^3$He (b). Plume dFe/xs$^3$He$_{Mn}$ (with xs$^3$He derived from dMn) (c) and TDFe/xs$^3$He$_{Mn}$ (d) at each station plotted against vent endmember fluid values. Note the logarithmic y-axis on (b) and (d) to make it easier to see the position of the points. The grey dashed line (b and d) shows a ratio of 1:1 where plume ratios are equal to vent ratios. Error bars represent the standard deviation of the average endmembers taken from multiple individual vents at each site (Table 1). Vent xs$^3$He data is from (Jean-Baptiste et al., 2004) and Fe concentration data from Beaulieu et al. (2013b) and Findlay et al. (2015) for TAG ($n$ = 6) and Rainbow ($n$ = 3), Chavagnac et al. (2018) and Pester et al. (2012) for Lucky Strike (average $n$ = 26) and Koschinsky et al. (2020) Beaulieu et al. (2013b) for Menez Gwen (average $n$ = 10).

The majority of Fe bearing particles in the unfiltered TDFe samples will be Fe-oxyhydroxides that will dissolve at pH 1.8 after >1 year storage. A possible exception being that a fraction of the particulates will be FeS$_2$ which may not dissolve with addition of HCl (German et al., 1991; Gartman et al., 2014). Extrapolation of experimental data of FeS$_2$ oxidation rates in seawater and acidic solution (Gartman and Luther, 2014; Constantin and Chiriță, 2013) indicate that 80-100 % of any FeS$_2$ present in our samples should have oxidised during the ~1 year between sampling and analysis. We anticipate that any FeS$_2$




present in our samples dissolved during sample storage and is included in our measured dFe concentrations. In this way, the plume TDFe/xs$^3$He and dMn/xs$^3$He ratios can be compared to vent dFe/xs$^3$He and dMn/xs$^3$He ratios, to examine whether the samples from a given cast capture the full extent of the plume, rather than just the margins. TDFe/xs$^3$He$_{Mn}$ and dMn/xs$^3$He values less than vent fluid values would indicate rapid particle formation and settling has removed Fe and/or Mn from the early plume.

The similarity between vent dFe/xs$^3$He and plume TDFe/xs$^3$He$_{Mn}$ at 0 km indicates minimal Fe has been lost from particle settling in the immediate plume over each vent site and that the plume cores were indeed sampled from the trace metal rosette (Fig. 4D). Similar results are apparent when comparing vent dMn/xs$^3$He and plume dMn/xs$^3$He from the standard rosette (Fig. 4B). One station at Rainbow shows TDFe/xs$^3$He$_{Mn}$ higher than the 1:1 vent plume ratio (Fig. 4D), however, this plume signal was located within 50 m of vents on the seafloor (2300 m) and therefore likely represents the narrow (usually

<1 to several metres wide) buoyant rising plume (see Table 1 caption and Fig. S8). In this instance its plausible that resuspended benthic Fe entrained in the rising plume near the seafloor may elevate the TDFe/xs$^3$He$_{Mn}$ to values higher than the vent dFe/xs$^3$He. Alternatively, the range of TDFe/xs$^3$He at the different sites may represent the combined uncertainty from the dual-Mn integration method.

Values of plume dMn/xs$^3$He fall marginally below the 1:1 ratio line for Lucky Strike and Rainbow (Fig. 4B) and could be a
result of lower sampling resolution on the standard rosette (Fig. S3 and S4), uncertainties associated with the plume integration, vent fluid endmembers or enhanced removal of dMn by particulates at Lucky Strike and Rainbow. In the case of Rainbow this could be the result of a higher particulate Fe-oxyhydroxide concentration in the plume due to the significantly higher Fe/H$_2$S ratio of the vent fluids (Fe/H$_2$S = 26), however, that is less likely to be the case for Lucky Strike where vent Fe/H$_2$S is lower (Fe/H$_2$S = 0.18) and sulphide concentrations are enough to potentially consume all the Fe.

All vents on this section of the North Mid-Atlantic Ridge have similar endmember xs$^3$He concentrations (18 ± 6 pmol/kg, $n$ = 4 vents (Jean-Baptiste et al., 2004), Table 1). Therefore, differences in vent dFe/xs$^3$He are solely driven by differences in Fe concentration, which range from 16 to 24100 µmol/kg across the four vent sites (Beaulieu et al., 2013b; Chavagnac et al., 2018; Koschinsky et al., 2020) and is determined by the geochemistry of each vent site (Table 1). The endmember vent fluids at Rainbow have the highest dFe/xs$^3$He (964 nmol/fmol at Rainbow in contrast to 278 nmol/fmol at TAG) as the
higher temperature, higher Cl$^-$ and low pH of fluids leach more Fe (along with other metals and rare earth elements) from the host rock in comparison to fluids circulating through the sites with basaltic rocks (Douville et al., 2002). Therefore, site by site differences in the vent fluid endmember dFe/xs$^3$He ratio was not simply related to the plume dFe/xs$^3$He$_{Mn}$. For instance, the highest dFe/xs$^3$He$_{Mn}$ plume ratio was observed over TAG (86 nmole/fmole), which was double that of the highest dFe/xs$^3$He$_{Mn}$ ratio over Rainbow (44 nmole/fmole) (Table 1, Fig. 4C), despite the 5-fold greater Fe content of Rainbow
fluids. In contrast, the TDFe/xs$^3$He$_{Mn}$ values were correlated with the vent fluid endmembers across our sites (Fig. 4D).



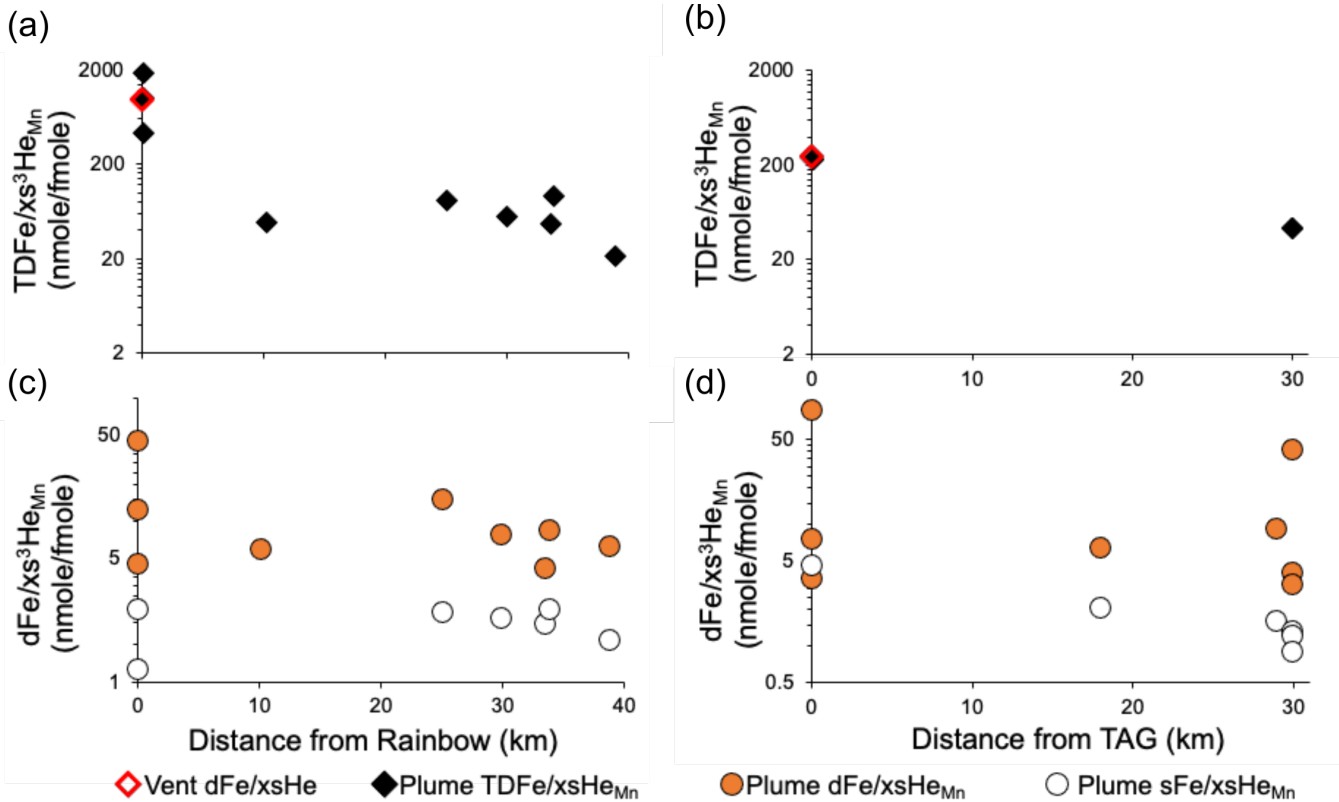

Figure 5. Plume TDFe/xs$^3$He$_{Mn}$ with distance (within 40km) from Rainbow (a) and TAG (within 30 km) (b). Change in plume dFe/xs$^3$He$_{Mn}$ and sFe/xs$^3$He$_{Mn}$ at Rainbow (c) and TAG (d). Note log scales on y-axis.

### 3.3 Dissolved – Particulate dynamics around the Rainbow and TAG sites

Over short spatial scales, a decrease in the Fe/xs$^3$He$_{Mn}$ ratio was seen across both the total and dissolved Fe size fractions, relative to the maximum ratios sampled over the vent sites (Fig. 5). Within 40km of both the Rainbow and TAG sites, TDFe/xs$^3$He$_{Mn}$ decreased significantly from the values over the vent site and those of the vent fluids. By comparing TDFe/xs$^3$He$_{Mn}$ of the 10-40 km stations with the dFe/xs$^3$He of the vent fluids, the amount of vent derived dFe that has been removed by formation and settling of particles can be estimated. This is ~94 and ~83 % for the Rainbow and TAG sites respectively. The dFe/xs$^3$He$_{Mn}$ ratios of the 10-40 km sites decreased to between 4 and 15 nmol/fmol (except for one station, 30 km west of TAG, which had a ratio of 38 nmol/fmol, Fig. 5D) in comparison to maximum values over the vent sites at 0 km. This indicates the importance of Fe removal by precipitation of particulates and the subsequent settling of large and/or heavy particulates within 10-20 km of the vent source at both sites, on time scales of days to weeks, leading to a smaller range of dFe/xs$^3$He$_{Mn}$ that may represent broader transport to the ocean interior.

It is particularly notable that despite observing an order of magnitude greater TDFe concentrations at Rainbow, relative to TAG (Fig. 5 and 4D), the dFe/xs$^3$He$_{Mn}$ values within 40 km are very similar across all size fractions for both sites. One exception is a station 30 km west of TAG, where a dFe/xs$^3$He$_{Mn}$ of 38 nmol/fmol is consistent with a younger plume signal





with less time for particle formation and settling. It is unlikely that there is a new hydrothermal source adding additional Fe, given how extensively this area of the MAR has been surveyed (Kinoshita et al., 1998). Experimental data examining the oxidation kinetics of Fe at the same stations around TAG found that samples from this station West of TAG had anomalous log K values, possibly as a result of interactions with organic matter (González-Santana et al., 2021). This could explain the
anomalously high dFe/xs$^3$He$_{Mn}$ at this site, however we are unable to explain why this site would have higher concentrations of organic matter or why dFe would interact differently with organic matter at this station compared to the other sites.

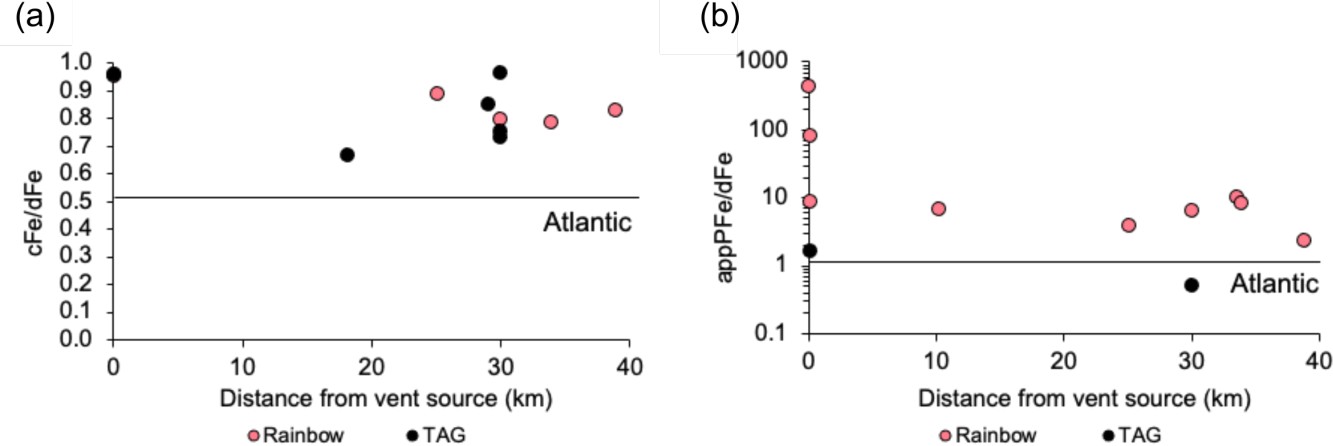

**Figure 6. Change in integrated cFe/dFe where cFe = dFe – sFe (a) and appPFe/dFe of the plume with distance from Rainbow and TAG where appPFe = TDFe – dFe (b). Note the log scale on the y-axis of (b). Solid black lines shows the same ratios for open**
**Atlantic deep waters taken from Milne et al. (2017) for appPFe/dFe and Kunde et al. (2019b) for cFe/dFe.**

The average cFe/dFe ratio of the TAG plume ($0.82 \pm 0.12$, $n = 6$) is similar to the Rainbow plume ($0.84 \pm 0.10$, $n = 7$), indicating that dFe is predominantly colloidal within 40 km of the vent source at both sites, with little change over these distances (Fig. 6A). In contrast, there is a clear difference in the appPFe/dFe ratio (Fig. 6B), which is elevated in the Rainbow plume and consistent with the higher concentration Fe source at this site. This highlights a substantially larger role
for dissolved-particulate interactions at the Rainbow site, relative to TAG. It is due to these strong particulate-dissolved exchanges that the dFe/xs$^3$He$_{Mn}$ ratios 40 km from source are largely similar for both TAG and Rainbow, despite the Rainbow neutrally buoyant plume having initially half the dFe/xs$^3$He over the vent site (44 nmol/fmol) relative to TAG (86 nmol/fmol) (Fig. 5C and D). If dFe was behaving entirely conservatively we would anticipate the range of dFe/xs$^3$He at the 10-40 km stations around Rainbow (4-15 nmol/fmol) to be lower than that of the equivalent sites around TAG (4-9
fmol/nmol, excluding the anomalous western station). The similarity in these ranges suggests the dFe/xs$^3$He ratio in the Rainbow plume was buffered by disaggregating or dissolving particles. Specifically, particles which were too large to enter the dissolved phase but not large or dense enough to form particles that rapidly settle out of the plume within the 40 km sampling area.

Separating whether particle disaggregation or dissolution is the driving mechanism behind the buffering of dFe
concentrations between the 0 km and 40 km stations at Rainbow is difficult from these results. We would anticipate particle



dissolution would transfer appPFe to the sFe fraction as aqueous ions. This would cause a shift in the cFe/dFe ratio (Fig. 6) and we did not observe this. This means that either disaggregation is the driving force of appPFe dFe exchange and particle dissolution is negligible, or there is dissolution followed by rapid re-precipitation as inorganic colloidal Fe or binding to colloidal sized Fe ligands.

**4. Wider implications**

In general, the large differences in appPFe-dFe dynamics over short spatial scales between the TAG and Rainbow plumes highlight the potential importance of particulate-dissolved Fe exchange in governing hydrothermal Fe transport from distinct vent sources. If particulate Fe concentrations extend the longevity of hydrothermal Fe in the deep ocean as hypothesised (Fitzsimmons et al., 2017) then it is likely that the residence time of hydrothermal Fe will depend on the amount of Fe released from vents on the seafloor and the presence of ligands to facilitate particulate-dissolved Fe exchange. In this conceptual model, we would anticipate the residence time of dFe in the Rainbow plume to be longer than the TAG plume, due to higher particulate Fe concentrations facilitating transport over longer distances. Alternatively, if hydrothermal dFe behaves conservatively (Resing et al., 2015) (and dissolved-particulate Fe exchange is negligible) then the longevity of hydrothermal Fe will depend on how much Fe forms colloidal and soluble phases in the early stages of the plume. In this instance, we would anticipate dFe in the TAG plume to have a longer residence time in the water column due to initially higher dFe/xs$^3$He (Fig. 4C and Table 1). In both conceptual models, Fe binding ligands are likely to play a key role by either facilitating particulate-dissolved Fe exchange or by stabilising dFe species. However, if the formation of colloids in the first several meters of plume rise is largely a result of inorganic precipitation of dFe then the role of ligands will be less important in the conservative colloids model. The "conservative Fe colloids" model has been tested in global biogeochemical models (Resing et al., 2015) and estimates a global hydrothermal Fe flux of $4 \pm 1$ Gmol yr$^{-1}$. A version of the "particulate-dissolved Fe exchange" model has also been tested and this produces a much lower flux estimate of $0.12 \pm 0.07$ Gmol yr$^{-1}$ (Roshan et al., 2020). Refining these models and deciding which is closer to the truth requires experimental data on the rate of particulate and dissolved Fe exchange and stability of colloidal phases in both hydrothermal plumes and the deep ocean.

Our results do not show constant linear trends between dFe and xs$^3$He (Fig. 5) that have been observed over larger distances in basin scale sampling efforts in the Pacific (Resing et al., 2015). As well as looking at integrated values we examined the correlation between xs$^3$He$_{Mn}$ and dFe across individual samples and found $r^2$ values of 0.956 to 0.811 for Lucky Strike and Menez Gwen, and 0.442 and 0.587 for Rainbow and TAG, respectively (Fig. S11). Interestingly, sites like TAG and Rainbow with higher particle concentrations had lower $r^2$ values. These deviations from linear relationships likely indicate ongoing particulate dissolved Fe exchange in the near field, which has been observed previously at smaller scale sampling resolution (Lough et al., 2019a).





## 4.1 What controls near-field dissolved Fe to Helium ratios

Given the significantly higher vent fluid $dFe/xs^3He$ at Rainbow compared with TAG (Table 1), the $dFe/xs^3He$ at distances 10 to 40 km from the vent sites is remarkably similar ($8 \pm 4$ versus $12 \pm 14$ nmol/fmol, repectively, $n = 5$). This would suggest there is a cap on the amount of vent fluid dFe that can be converted into dFe in the plume, which is possibly the result of
similar Fe binding ligand concentrations and/or strength to at both sites. This could be in the form of weakly binding pervasive background Fe binding ligands present in deep ocean waters or similarities in Fe binding ligands sourced locally from the ecosystems of both vents (Kleint et al., 2016). Alternatively, $Fe^{2+}$ oxidation rates may determine the extent to which Fe is separated between dissolved and particulate phases and hence the $dFe/xs^3He$. Rates of $Fe^{2+}$ oxidation measured at both these sites show a similar range of $Fe^{2+}$ half-lives in the plumes (TAG = 1-130 mins, Rainbow = 20-160 mins (González-
Santana et al., 2021)). The initial concentration of Fe released from Rainbow vents is five times higher than at TAG, despite a similar He concentration, and if $Fe^{2+}$ oxidation rates were the main driver of dFe concentrations, the Rainbow plume should have higher ratios of $dFe/xs^3He$. This contrasts with our observations, suggesting that $Fe^{2+}$ oxidation rates are less important in establishing plume $dFe/xs^3He$ at the scales of 10s of km than ligand strength, concentration, or inorganic colloid formation.

Vent fluids with molar $Fe/H_2S >1$ are likely to precipitate higher concentrations of $FeS_2$ nanoparticles in plumes (Gartman et al., 2014). Given that TAG has a lower $Fe/H_2S$ ratio than Rainbow ($Fe/H_2S$ is 1 mmol/mmol for TAG vents and 26 mmol/mmol for Rainbow (Table 1)), we should anticipate a higher concentration of $FeS_2$ nanoparticles in the TAG plume. Higher concentrations of $FeS_2$ nanoparticles may offer an explanation as to why the maximum $dFe/xs^3He$ of the TAG plume (86 fmol/nmol) was high in comparison to Rainbow (44 fmol/nmol). The 2-fold difference in maximum plume $dFe/xs^3He$
between TAG and Rainbow is small compared to the 18-fold difference in vent fluid $Fe/H_2S$ (Table 1) and based on the observed trend between $FeS_2$ nanoparticle concentration and $Fe/H_2S$ shown in Gartman et al. (2014) we would only anticipate an additional 4 % $FeS_2$ concentration in the dissolved phase at TAG. Suggesting that the formation of $FeS_2$ nanoparticles in the nascent plume cannot fully explain differences in plume $dFe/xs^3He$ between sites.  From our comparison, it would seem that for vent sites located along the same ridge, ligand concentration and strength are likely to be
a more important control on near-field $dFe/xs^3He$ than vent fluid chemistry and $Fe^{2+}$ oxidation rate (set by water column $O_2$ and pH (Santana-Casiano et al., 2000; Millero et al., 1987)).

If we only consider the dFe flux from plumes, excluding any contribution from particulates, site-to-site differences in the chemistry of hydrothermal systems give a range of 8-12 nmol/fmol (average range from Rainbow and TAG at 10-40 km) for $dFe/xs^3He$ in the near-field (Fig. 5). Therefore, any subsequent dFe flux calculated based on $dFe/xs^3He$ will only vary by a
similar magnitude. This would mean that current biogeochemical models using a global $dFe/xs^3He$ ratio of 10 nmol/fmol (within the 8-12 nmol/fmol range observed at 10-50 km) are using a reasonably well-defined input flux of "dissolved" hydrothermal Fe into the deep ocean. However, given the differences between sites in $appFe/dFe$ (Fig. 6) we need to consider the impact of particulate Fe on the hydrothermal Fe flux and how this may vary across the global ocean ridge crest.



## 4.2 Particle plumes escaping the mid-ocean ridge valley

All stations maintain a cFe/dFe ratio greater than open ocean N. Atlantic cFe/dFe values of 0.5 (Fig. 6A). Showing that within the ridge valley, plume waters maintain a higher colloidal load than open ocean N. Atlantic waters. If the residence time of Fe rich plumes trapped within the Mid-Atlantic ridge valley are similar to that estimated from models of dispersion (Vic et al., 2018), then wherever deep waters escape the ridge valley they may also carry elevated concentrations of particulate and dissolved Fe to the deep Atlantic ocean. Using Stokes' law we can calculate if it is reasonable to expect

plume waters exiting the ridge valley to carry sufficient particulate Fe or whether it will have settled out to the sediment. Plume waters from Lucky Strike take ~30 days to exit the ridge valley based on lagrangian particle dispersion experiments (Vic et al., 2018). As there are no other comparative dispersion experiments for other sites on the MAR, we can assume that the dispersion time for plume waters to exit the ridge valley is the same for other vents on the MAR as it is at Lucky Strike. The approximate distance from sources of venting at the centre of the valley to the outer ridge flank is ~100 km. We can then

use Stokes' law to calculate that a particle of ferrihydrite 20 μm in size (an upper limit on Fe particle size based on images (Feely et al., 1994; Breier et al., 2014; Lough et al., 2019b; Lough et al., 2019a; Lough et al., 2017; Toner et al., 2009; Toner et al., 2016)) would have settled 1.5 m through the water column as plume waters travel from the vent source out of the ridge valley. This is not enough for particulates to reach the seafloor given buoyant plumes inject particulates 100's of metres into the water column. This is a maximum estimate of settling when we consider that 1) Fe rich particles sampled within plumes

are usually much smaller than 20 μm in size (average of $6 \pm 6$ μm, $n = 28$, based on published images (Feely et al., 1994; Breier et al., 2014; Lough et al., 2019b; Lough et al., 2019a; Lough et al., 2017; Toner et al., 2009; Toner et al., 2016)), 2) Particles are often a mix of Fe oxyhydroxide minerals and organic carbon making them less dense than pure ferrihydrite (Toner et al., 2009; Yucel et al., 2011; Gartman et al., 2014; Lough et al., 2017; Hoffman et al., 2018; Lough et al., 2019b; Lough et al., 2019a), and 3) lagrangian particles in dispersion models are carried ~100 m vertically beyond the neutrally

buoyant plume depth by turbulent mixing across the ridge (Vic et al., 2018) and tidal forces can shift neutrally buoyant plume depths by ~100 m within the ridge valley (Jean-Baptiste et al., 2004). This would counteract the effects of particle settling and we see some evidence of this in Fig. 3 where the maximum concentration of elements is at a shallower depth (~1900 m) at stations away from the vent source (Fig. 3D and E) in comparison to stations over the vents (2000-2300 m, Fig. 3A and B). It is therefore likely that the TDFe concentrations we observe within 10-40 km of the vents will be similar to

those of waters exiting the ridge valley and entering the deep ocean. This creates the idea of a "leaky ridge" rather than a "leaky vent" where Fe enriched ridge valley waters escape from the ridge at key fracture zones (Toner et al., 2009). Hence, for slow spreading mid-ocean ridges (*i.e.* ridges with a substantially deep ridge valley), it is the point at which plume waters exit the ridge valley, the Fe carrying capacity of these water (particulates as well as dissolved phase), and the rate at which Fe is removed from the plume as settling particles, that is key to determining the impact of hydrothermal vents on deep

ocean Fe concentrations. This highlights the importance of constraining the physical mixing regimes of waters moving through ridge valleys, especially for slow spreading ridges where plumes will initially be topographically constrained,





moving away from a two dimensional interpretation of basin scale plumes defined by the path of a ship across the ocean (Nishioka et al., 2013; Saito et al., 2013; Resing et al., 2015; Fitzsimmons et al., 2017).

It is possible for global biogeochemical models to overestimate Fe fluxes from some mid-ocean ridges, as the complex, mesoscale mixing regimes through ridge valleys are not parameterised at the global scale. In order for this to be the case, the pathways that ridge valley waters travel from the vent source to exiting the ridge valley would have to be significantly longer than those modelled for Lucky Strike (Vic et al., 2018), allowing for an extended period of time for scavenging and precipitation of Fe (resulting in lower $dFe/xs^3He$ values than those used in models and observed in this study) leading to lower Fe fluxes. If we lead with our assumption that plume dispersion time at Lucky Strike (Vic et al., 2018) is

representative of most plumes emanating from Mid-Ocean Ridges, (given that the physical mixing regimes acting on the plume and topographic controls will be similar along the ridge) and that particulates at the 10-40 km distance are likely to remain suspended in the water column or re-dissolve, then using $TDFe/xs^3He$ ratios of $55 \pm 24$ *n = 7* (average all TAG and Rainbow stations at 10-40 km distance) in biogeochemical models maybe more representative of the hydrothermal Fe flux which is 5x greater than the ratio of 10 used currently (Tagliabue, 2010; Resing et al., 2015).

**4.3 Future Work**

Similar process studies that sample plumes from vents with different geochemistry along a ridge will be needed to test the ideas discussed here further. Looking specifically at the dispersion of plumes through the ridge valley and whether there is any difference in Fe binding ligand strength/concentration between vent sites with different Fe concentrations and different amounts of diffuse flow that likely act as a source of ligands. Our findings from the comparison of TAG and Rainbow show

that the inorganic geochemistry of individual vents sites plays a minor role in dictating the near-field plume dFe concentration, however, an excess of plume pFe resulting from higher vent fluid Fe concentrations may support dFe longevity via dFe-pFe exchange provided the particulates remain suspended, disaggregate or re-dissolve in the water column. It is therefore possible that vents situated in an ultramafic setting (*i.e.* low pH and high $Cl^-$ content due to interaction with ultramafic rocks during hydrothermal circulation) may provide more Fe to the deep ocean in comparison to basalt

hosted systems. However, constraining the full impact of ultramafic vent sites on the net global hydrothermal Fe flux requires more knowledge about the frequency of their occurrence along the global ridge crest (Baker et al., 2016).

To improve our estimates of how much hydrothermal Fe fertilizes surface ocean productivity we need further information on, the location and frequency of vent systems along the global ridge crest, how much variability there is in the hydrothermal ligand source between vent sites, what controls the rate of particulate dissolved exchange in the plume, and how rapidly

hydrothermal Fe is scavenged from the deep ocean?



## 5. Conclusions

Our field results show that care must be taken when extrapolating Fe/xs$^3$He results from ocean survey sampling, e.g. as part of GEOTRACES. This is due to uncertainties associated with the at-sea sampling strategy and the temporal nature of plume dynamics that can yield significant variability in Fe/xs$^3$He ratios. Future studies may wish to explore technical solutions to

this issue by finding a way to sample He and trace metals from the same cast. Two potential solutions could be either sampling with a remote operated vehicle in addition to the rosette system or adding an additional tap to sampling bottles with a valve that prevents back flow from the Cu pipes used to sample He, the downside to this approach being that there would be limited water volume for sampling trace metals and their isotopes after He sampling. We recommend that measurements of dMn across different sampling systems are a useful means by which to minimise sampling uncertainties, especially when

combined with ORP and LSS sensors to target plume sampling across different deployments. Ultimately, when dMn and xs$^3$He measurements are used alongside both dFe and TDFe observations it is possible to link the observed plume dynamics to vent fluid endmembers and determine the important dissolved – particulate dynamics that shape the Fe/xs$^3$He signals that integrate at broader spatial scales.

Despite 5x higher concentrations of Fe (but similar xs$^3$He) venting from Rainbow relative to TAG we observed lower

dFe/xs$^3$He in the Rainbow plume over the vent site (Fig. 4C). The additional Fe venting at Rainbow was converted into particulates and this was reflected in the ratio of appPFe/dFe at Rainbow which is higher than TAG (Fig. 6B). These results suggest that there is a threshold placed on the amount of venting dFe that can be converted to plume dFe in the near field, most likely by the concentration of Fe binding ligands. Greater than 80 % of vent fluid Fe formed large dense particulates that settled rapidly within 10 km's and a smaller but significant fraction remained suspended in the water column. The

persisting particulate Fe may enhance the longevity of plume dFe through particulate dissolved exchange. The extent of ongoing particulate dissolved Fe exchange with further plume dispersion will depend on the speciation and size distribution of particulates as well as ligand strength, concentration and longevity which may differ between these two sites. Future work examining any differences in particle speciation and size between these two sites will be better placed to determine whether the higher concentration of suspended particulates at Rainbow will enhance the longevity of the hydrothermal plume relative

to the TAG plume.

Based on our results it is likely that Fe binding ligand concentrations place a limit on the amount of dFe released from the vents that is converted into plume dFe. Given the variability in our data it is difficult to say definitively whether or not this dFe limit is different between the vent sites as a result of ligand supply from local ecosystems or if it is similar as a result of ligand supply from ubiquitous deep ocean waters.

**Author Contributions**

AT and ML wrote the proposal to secure funding for the project. The sampling design for fieldwork was conceived by AT, ML and JR. AJML, AT, JR, NW and ML mobilised equipment and consumables for fieldwork. Samples were collected in



the field by AJML, AT, JR and ML. All sample analysis was conducted by AL and CD apart from Helium analysis which

was paid for at Woods Hole Oceanographic Institute. NW calculated xs$^3$He values. AJML analysed and interpreted the data,

produced the figures, and wrote the manuscript. All authors provided comments on subsequent drafts of the manuscript.

**Acknowledgements**

This work is part of the FRidge project (GEOTRACES section GA13) which was supported by the Natural Environment

Research Council funding (NERC United Kingdom Grants NE/N010396/1 to MCL and NE/N009525/1 to AT). We thank

Shaun Rigby and Ric Williams for Helium sampling at sea and Bill Jenkins for He analysis. We also thank the Captain and

crew of the *RRS James Cook* and everyone that contributed to the GA13 sampling effort on board. CD was additionally

funded by the Occitanie Pyrenees-Mediterranée Regional Council. JR was funded by NOAA Ocean Exploration and Earth

Interactions programs from PMEL and CICOES. The data from the GA13 transect is available (to view and download) as

part of the GEOTRACES international data product which can be accessed online via https://www.egeotraces.org/.

References,

Ardyna, M., Lacour, L., Sergi, S., d'Ovidio, F., Sallée, J.-B., Rembauville, M., Blain, S., Tagliabue, A., Schlitzer, R., Jeandel, C., Arrigo, K. R., and Claustre, H.: Hydrothermal vents trigger massive phytoplankton blooms in the Southern Ocean, Nature communications, 10, 2451, 10.1038/s41467-019-09973-6, 2019.

Baker, E. T., Resing, J. A., Haymon, R. M., Tunnicliffe, V., Lavelle, J. W., Martinez, F., Ferrini, V., Walker, S. L., and Nakamura, K.:
How many vent fields? New estimates of vent field populations on ocean ridges from precise mapping of hydrothermal discharge locations, Earth and Planetary Science Letters, 449, 186-196, http://dx.doi.org/10.1016/j.epsl.2016.05.031, 2016.

Beaulieu, S. E., Baker, E. T., German, C. R., and Maffei, A.: An authoritative global database for active submarine hydrothermal vent fields, Geochemistry, Geophysics, Geosystems, 14, 4892-4905, 10.1002/2013gc004998, 2013a.

Beaulieu, S. E., Baker, E. T., German, C. R., and Maffei, A.: An authoritative global database for active submarine hydrothermal vent
fields, Geochemistry Geophysics Geosystems, 14, 4892-4905, 10.1002/2013gc004998, 2013b.

Bennett, S. A., Achterberg, E. P., Connelly, D. P., Statham, P. J., Fones, G. R., and German, C. R.: The distribution and stabilisation of dissolved Fe in deep-sea hydrothermal plumes, Earth and Planetary Science Letters, 270, 157-167, 10.1016/j.epsl.2008.01.048, 2008.

Breier, J. A., Sheik, C. S., Gomez-Ibanez, D., Sayre-McCord, R. T., Sanger, R., Rauch, C., Coleman, M., Bennett, S. A., Cron, B. R., Li, M., German, C. R., Toner, B. M., and Dick, G. J.: A large volume particulate and water multi-sampler with in situ preservation for
microbial and biogeochemical studies, Deep-Sea Res. Part I-Oceanogr. Res. Pap., 94, 195-206, 10.1016/j.dsr.2014.08.008, 2014.

Chavagnac, V., Leleu, T., Fontaine, F., Cannat, M., Ceuleneer, G., and Castillo, A.: Spatial Variations in Vent Chemistry at the Lucky Strike Hydrothermal Field, Mid-Atlantic Ridge (37°N): Updates for Subseafloor Flow Geometry From the Newly Discovered Capelinhos Vent, Geochemistry, Geophysics, Geosystems, 19, 4444-4458, 10.1029/2018gc007765, 2018.

Constantin, C. A. and Chiriţă, P.: Oxidative dissolution of pyrite in acidic media, Journal of Applied Electrochemistry, 43, 659-666,
10.1007/s10800-013-0557-y, 2013.

Cowen, J. P., Massoth, G. J., and Baker, E. T.: Bacterial scavenging of Mn and Fe in a Mid Field to Far Field hydrothermal particle plume, Nature, 322, 169-171, 10.1038/322169a0, 1986.

Cowen, J. P., Massoth, G. J., and Feely, R. A.: Scavenging rates of dissolved manganese in a hydrothermal vent plume, Deep Sea Research Part A. Oceanographic Research Papers, 37, 1619-1637, http://dx.doi.org/10.1016/0198-0149(90)90065-4, 1990.
Sampling and Sample-handling Protocols for GEOTRACES Cruises, last access: 16/12/19.

Douville, E., Charlou, J. L., Oelkers, E. H., Bienvenu, P., Colon, C. F. J., Donval, J. P., Fouquet, Y., Prieur, D., and Appriou, P.: The rainbow vent fluids (36 degrees 14 ' N, MAR): the influence of ultramafic rocks and phase separation on trace metal content in Mid-Atlantic Ridge hydrothermal fluids, Chem. Geol., 184, 37-48, 10.1016/s0009-2541(01)00351-5, 2002.

Feely, R. A., Gendron, J. F., Baker, E. T., and Lebon, G. T.: Hydrothermal plumes along the east pacific rise, 8-degrees-40' to 11-degrees-
50'N - particle distribution and composition, Earth and Planetary Science Letters, 128, 19-36, 10.1016/0012-821x(94)90023-x, 1994.

Field, M. P. and Sherrell, R. M.: Dissolved and particulate Fe in a hydrothermal plume at 9 degrees 45 ' N, East Pacific Rise: Slow Fe (II) oxidation kinetics in Pacific plumes, Geochimica et Cosmochimica Acta, 64, 619-628, 10.1016/s0016-7037(99)00333-6, 2000.





Findlay, A. J., Gartman, A., Shaw, T. J., and Luther, G. W.: Trace metal concentration and partitioning in the first 1.5 m of hydrothermal vent plumes along the Mid-Atlantic Ridge: TAG, Snakepit, and Rainbow, Chem. Geol., 412, 117-131, 10.1016/j.chemgeo.2015.07.021, 2015.

Fitzsimmons, J. N., John, S. G., Marsay, C. M., Hoffman, C. L., Nicholas, Sarah L., Toner, B. M., German, C. R., and Sherrell, R. M.: Iron persistence in a distal hydrothermal plume supported by dissolved–particulate exchange, Nat. Geosci., 10, 195, 10.1038/ngeo2900 https://www.nature.com/articles/ngeo2900#supplementary-information, 2017.

Gartman, A. and Luther, G. W.: Oxidation of synthesized sub-micron pyrite (FeS2) in seawater, Geochimica Et Cosmochimica Acta, 144, 96-108, 10.1016/j.gca.2014.08.022, 2014.

Gartman, A., Findlay, A. J., and Luther, G. W.: Nanoparticulate pyrite and other nanoparticles are a widespread component of hydrothermal vent black smoker emissions, Chem. Geol., 366, 32-41, 10.1016/j.chemgeo.2013.12.013, 2014.

German, C. R., Campbell, A. C., and Edmond, J. M.: Hydrothermal scavenging at the Mid-Atlantic Ridge - modification of trace-element dissolved fluxes, Earth and Planetary Science Letters, 107, 101-114, 10.1016/0012-821x(91)90047-l, 1991.

González-Santana, D., González-Dávila, M., Lohan, M. C., Artigue, L., Planquette, H., Sarthou, G., Tagliabue, A., and Santana-Casiano, J. M.: Variability in iron (II) oxidation kinetics across diverse hydrothermal sites on the northern Mid Atlantic Ridge, Geochimica et Cosmochimica Acta, 297, 143-157, https://doi.org/10.1016/j.gca.2021.01.013, 2021.

Gottschalk, J., Battaglia, G., Fischer, H., Frölicher, T. L., Jaccard, S. L., Jeltsch-Thömmes, A., Joos, F., Köhler, P., Meissner, K. J., Menviel, L., Nehrbass-Ahles, C., Schmitt, J., Schmittner, A., Skinner, L. C., and Stocker, T. F.: Mechanisms of millennial-scale atmospheric CO2 change in numerical model simulations, Quaternary Science Reviews, 220, 30-74, https://doi.org/10.1016/j.quascirev.2019.05.013, 2019.

Hawkes, J. A., Connelly, D. P., Gledhill, M., and Achterberg, E. P.: The stabilisation and transportation of dissolved iron from high temperature hydrothermal vent systems, Earth and Planetary Science Letters, 375, 280-290, 10.1016/j.epsl.2013.05.047, 2013.

Hoffman, C. L., Nicholas, S. L., Ohnemus, D. C., Fitzsimmons, J. N., Sherrell, R. M., German, C. R., Heller, M. I., Lee, J.-m., Lam, P. J., and Toner, B. M.: Near-field iron and carbon chemistry of non-buoyant hydrothermal plume particles, Southern East Pacific Rise 15°S, Marine Chemistry, https://doi.org/10.1016/j.marchem.2018.01.011, 2018.

James, R. H. and Elderfield, H.: Dissolved and particulate trace metals in hydrothermal plumes at the Mid-Atlantic Ridge, Geophysical Research Letters, 23, 3499-3502, 10.1029/96gl01588, 1996.

Jean-Baptiste, P., Fourré, E., Charlou, J.-L., German, C. R., and Radford-Knoery, J.: Helium isotopes at the Rainbow hydrothermal site (Mid-Atlantic Ridge, 36°14′N), Earth and Planetary Science Letters, 221, 325-335, https://doi.org/10.1016/S0012-821X(04)00094-9, 2004.

Jenkins, W. J., Smethie, W. M., Boyle, E. A., and Cutter, G. A.: Water mass analysis for the U.S. GEOTRACES (GA03) North Atlantic sections, Deep Sea Research Part II: Topical Studies in Oceanography, 116, 6-20, https://doi.org/10.1016/j.dsr2.2014.11.018, 2015a.

Jenkins, W. J., Lott, D. E., Longworth, B. E., Curtice, J. M., and Cahill, K. L.: The distributions of helium isotopes and tritium along the U.S. GEOTRACES North Atlantic sections (GEOTRACES GAO3), Deep Sea Research Part II: Topical Studies in Oceanography, 116, 21-28, https://doi.org/10.1016/j.dsr2.2014.11.017, 2015b.

Kinoshita, M., Von Herzen, R. P., Matsubayashi, O., and Fujioka, K.: Tidally-driven effluent detected by long-term temperature monitoring at the TAG hydrothermal mound, Mid-Atlantic Ridge, Physics of the Earth and Planetary Interiors, 108, 143-154, https://doi.org/10.1016/S0031-9201(98)00092-2, 1998.

Kleint, C., Hawkes, J. A., Sander, S. G., and Koschinsky, A.: Voltammetric Investigation Of Hydrothermal Iron Speciation, Frontiers in Marine Science, 3, 10.3389/fmars.2016.00075, 2016.

Koschinsky, A., Schmidt, K., and Garbe-Schönberg, D.: Geochemical time series of hydrothermal fluids from the slow-spreading Mid-Atlantic Ridge: Implications of medium-term stability, Chem. Geol., 119760, https://doi.org/10.1016/j.chemgeo.2020.119760, 2020.

Kunde, K., Wyatt, N. J., Gonzalez-Santana, D., Tagliabue, A., Mahaffey, C., and Lohan, M. C.: Iron Distribution in the Subtropical North Atlantic: The Pivotal Role of Colloidal Iron, Glob. Biogeochem. Cycle, 16, 10.1029/2019gb006326, 2019a.

Kunde, K., Wyatt, N. J., González-Santana, D., Tagliabue, A., Mahaffey, C., and Lohan, M. C.: Iron Distribution in the Subtropical North Atlantic: The Pivotal Role of Colloidal Iron, Glob. Biogeochem. Cycle, 33, 1532-1547, https://doi.org/10.1029/2019GB006326, 2019b.

Lahaye, N., Gula, J., Thurnherr, A. M., Reverdin, G., Bouruet-Aubertot, P., and Roullet, G.: Deep Currents in the Rift Valley of the North Mid-Atlantic Ridge, Frontiers in Marine Science, 6, 17, 10.3389/fmars.2019.00597, 2019.

Lavelle, J. W., Cowen, J. P., and Massoth, G. J.: A model for the deposition of hydrothermal manganese near ridge crests, Journal of Geophysical Research: Oceans, 97, 7413-7427, https://doi.org/10.1029/92JC00406, 1992.

Lough, A. J. M., Homoky, W. B., Connelly, D. P., Comer-Warner, S. A., Nakamura, K., Abyaneh, M. K., Kaulich, B., and Mills, R. A.: Soluble iron conservation and colloidal iron dynamics in a hydrothermal plume, Chem. Geol., 511, 225-237, https://doi.org/10.1016/j.chemgeo.2019.01.001, 2019a.

Lough, A. J. M., Klar, J. K., Homoky, W. B., Comer-Warner, S. A., Milton, J. A., Connelly, D. P., James, R. H., and Mills, R. A.: Opposing authigenic controls on the isotopic signature of dissolved iron in hydrothermal plumes, Geochimica Et Cosmochimica Acta, 202, 1-20, 10.1016/j.gca.2016.12.022, 2017.



Lough, A. J. M., Connelly, D. P., Homoky, W. B., Hawkes, J. A., Chavagnac, V., Castillo, A., Kazemian, M., Nakamura, K., Araki, T., Kaulich, B., and Mills, R. A.: Diffuse Hydrothermal Venting: A Hidden Source of Iron to the Oceans, Frontiers in Marine Science, 6, 14, 10.3389/fmars.2019.00329, 2019b.

Lupton, J. E. and Craig, H.: A Major Helium-3 Source at 15°S on the East Pacific Rise, Science, 214, 13-18, 10.1126/science.214.4516.13, 1981.

Lupton, J. E., Weiss, R. F., and Craig, H.: Mantle helium in hydrothermal plumes in the Galapagos Rift, Nature, 267, 603-604, 10.1038/267603a0, 1977.

Massoth, G. J., Baker, E. T., Lupton, J. E., Feely, R. A., Butterfield, D. A., Von Damm, K. L., Roe, K. K., and Lebon, G. T.: Temporal and spatial variability of hydrothermal manganese and iron at Cleft segment, Juan de Fuca Ridge, Journal of Geophysical Research: Solid Earth, 99, 4905-4923, https://doi.org/10.1029/93JB02799, 1994.

Millero, F. J., Sotolongo, S., and Izaguirre, M.: The Oxidation Kinetics of Fe(II) in Seawater, Geochimica et Cosmochimica Acta, 51, 793-801, 10.1016/0016-7037(87)90093-7, 1987.

Milne, A., Schlosser, C., Wake, B. D., Achterberg, E. P., Chance, R., Baker, A. R., Forryan, A., and Lohan, M. C.: Particulate phases are key in controlling dissolved iron concentrations in the (sub)tropical North Atlantic, Geophysical Research Letters, 44, 2377-2387, 10.1002/2016gl072314, 2017.

Mottl, M. J. and McConachy, T. F.: Chemical processes in buoyant hydrothermal plumes on the East Pacific Rise near 21ºN., Geochimica et Cosmochimica Acta, 54, 1911-1927, 1990.

Nishioka, J., Obata, H., and Tsumune, D.: Evidence of an extensive spread of hydrothermal dissolved iron in the Indian Ocean, Earth and Planetary Science Letters, 361, 26-33, 10.1016/j.epsl.2012.11.040, 2013.

Obata, H., Karatani, H., and Nakayama, E.: Automated determination of iron in seawater by chelating resin concentration and chemiluminescence detection, Anal. Chem., 65, 1524-1528, 10.1021/ac00059a007, 1993.

Pester, N. J., Reeves, E. P., Rough, M. E., Ding, K., Seewald, J. S., and Seyfried, W. E.: Subseafloor phase equilibria in high-temperature

hydrothermal fluids of the Lucky Strike Seamount (Mid-Atlantic Ridge, 37°17′N), Geochimica et Cosmochimica Acta, 90, 303-322, https://doi.org/10.1016/j.gca.2012.05.018, 2012.

Resing, J. A. and Mottl, M. J.: Determination of manganese in seawater using flow injection analysis with on-line preconcentration and spectrophotometric detection, Anal. Chem., 64, 2682-2687, 10.1021/ac00046a006, 1992.

Resing, J. A., Sedwick, P. N., German, C. R., Jenkins, W. J., Moffett, J. W., Sohst, B. M., and Tagliabue, A.: Basin-scale transport of

hydrothermal dissolved metals across the South Pacific Ocean, Nature, 523, 200-U140, 10.1038/nature14577, 2015.

Roshan, S., DeVries, T., Wu, J., John, S., and Weber, T.: Reversible scavenging traps hydrothermal iron in the deep ocean, Earth and Planetary Science Letters, 542, 116297, https://doi.org/10.1016/j.epsl.2020.116297, 2020.

Saito, M. A., Noble, A. E., Tagliabue, A., Goepfert, T. J., Lamborg, C. H., and Jenkins, W. J.: Slow-spreading submarine ridges in the South Atlantic as a significant oceanic iron source, Nat. Geosci., 6, 775-779, 10.1038/ngeo1893, 2013.

Santana-Casiano, J. M., González-Dávila, M., Rodríguez, M. J., and Millero, F. J.: The effect of organic compounds in the oxidation kinetics of Fe(II), Marine Chemistry, 70, 211-222, https://doi.org/10.1016/S0304-4203(00)00027-X, 2000.

Schine, C. M. S., Alderkamp, A.-C., van Dijken, G., Gerringa, L. J. A., Sergi, S., Laan, P., van Haren, H., van de Poll, W. H., and Arrigo, K. R.: Massive Southern Ocean phytoplankton bloom fed by iron of possible hydrothermal origin, Nature communications, 12, 1211, 10.1038/s41467-021-21339-5, 2021.

Schlitzer, R., Anderson, R. F., Dodas, E. M., Lohan, M., Geibert, W., Tagliabue, A., Bowie, A., Jeandel, C., Maldonado, M. T., Landing, W. M., Cockwell, D., Abadie, C., Abouchami, W., Achterberg, E. P., Agather, A., Aguliar-Islas, A., van Aken, H. M., Andersen, M., Archer, C., Auro, M., de Baar, H. J., Baars, O., Baker, A. R., Bakker, K., Basak, C., Baskaran, M., Bates, N. R., Bauch, D., van Beek, P., Behrens, M. K., Black, E., Bluhm, K., Bopp, L., Bouman, H., Bowman, K., Bown, J., Boyd, P., Boye, M., Boyle, E. A., Branellec, P., Bridgestock, L., Brissebrat, G., Browning, T., Bruland, K. W., Brumsack, H.-J., Brzezinski, M., Buck, C. S., Buck, K. N., Buesseler, K.,

Bull, A., Butler, E., Cai, P., Mor, P. C., Cardinal, D., Carlson, C., Carrasco, G., Casacuberta, N., Casciotti, K. L., Castrillejo, M., Chamizo, E., Chance, R., Charette, M. A., Chaves, J. E., Cheng, H., Chever, F., Christl, M., Church, T. M., Closset, I., Colman, A., Conway, T. M., Cossa, D., Croot, P., Cullen, J. T., Cutter, G. A., Daniels, C., Dehairs, F., Deng, F., Dieu, H. T., Duggan, B., Dulaquais, G., Dumousseaud, C., Echegoyen-Sanz, Y., Edwards, R. L., Ellwood, M., Fahrbach, E., Fitzsimmons, J. N., Russell Flegal, A., Fleisher, M. Q., van de Flierdt, T., Frank, M., Friedrich, J., Fripiat, F., Fröllje, H., Galer, S. J. G., Gamo, T., Ganeshram, R. S., Garcia-Orellana, J., Garcia-

Solsona, E., Gault-Ringold, M., George, E., Gerringa, L. J. A., Gilbert, M., Godoy, J. M., Goldstein, S. L., Gonzalez, S. R., Grissom, K., Hammerschmidt, C., Hartman, A., Hassler, C. S., Hathorne, E. C., Hatta, M., Hawco, N., Hayes, C. T., Heimbürger, L.-E., Helgoe, J., Heller, M., Henderson, G. M., Henderson, P. B., van Heuven, S., Ho, P., Horner, T. J., Hsieh, Y.-T., Huang, K.-F., Humphreys, M. P., Isshiki, K., Jacquot, J. E., Janssen, D. J., Jenkins, W. J., John, S., Jones, E. M., Jones, J. L., Kadko, D. C., Kayser, R., Kenna, T. C., Khondoker, R., Kim, T., Kipp, L., Klar, J. K., Klunder, M., Kretschmer, S., Kumamoto, Y., Laan, P., Labatut, M., Lacan, F., Lam, P. J.,

Lambelet, M., Lamborg, C. H., Le Moigne, F. A. C., Le Roy, E., Lechtenfeld, O. J., Lee, J.-M., Lherminier, P., Little, S., López-Lora, M., Lu, Y., Masque, P., Mawji, E., McClain, C. R., Measures, C., Mehic, S., Barraqueta, J.-L. M., van der Merwe, P., Middag, R., Mieruch, S., Milne, A., Minami, T., Moffett, J. W., Moncoiffe, G., Moore, W. S., Morris, P. J., Morton, P. L., Nakaguchi, Y., Nakayama, N., Niedermiller, J., Nishioka, J., Nishiuchi, A., Noble, A., Obata, H., Ober, S., Ohnemus, D. C., van Ooijen, J., O'Sullivan, J., Owens, S.,



Pahnke, K., Paul, M., Pavia, F., Pena, L. D., Peters, B., Planchon, F., Planquette, H., Pradoux, C., Puigcorbé, V., Quay, P., Queroue, F.,
Radic, A., Rauschenberg, S., Rehkämper, M., Rember, R., Remenyi, T., Resing, J. A., Rickli, J., Rigaud, S., Rijkenberg, M. J. A., Rintoul,
S., Robinson, L. F., Roca-Martí, M., Rodellas, V., Roeske, T., Rolison, J. M., Rosenberg, M., Roshan, S., Rutgers van der Loeff, M. M.,
Ryabenko, E., Saito, M. A., Salt, L. A., Sanial, V., Sarthou, G., Schallenberg, C., Schauer, U., Scher, H., Schlosser, C., Schnetger, B.,
Scott, P., Sedwick, P. N., Semiletov, I., Shelley, R., Sherrell, R. M., Shiller, A. M., Sigman, D. M., Singh, S. K., Slagter, H. A., Slater, E.,
Smethie, W. M., Snaith, H., Sohrin, Y., Sohst, B., Sonke, J. E., Speich, S., Steinfeldt, R., Stewart, G., Stichel, T., Stirling, C. H., Stutsman,
J., Swarr, G. J., Swift, J. H., Thomas, A., Thorne, K., Till, C. P., Till, R., Townsend, A. T., Townsend, E., Tuerena, R., Twining, B. S.,
Vance, D., Velazquez, S., Venchiarutti, C., Villa-Alfageme, M., Vivancos, S. M., Voelker, A. H. L., Wake, B., Warner, M. J., Watson, R.,
van Weerlee, E., Alexandra Weigand, M., Weinstein, Y., Weiss, D., Wisotzki, A., Woodward, E. M. S., Wu, J., Wu, Y., Wuttig, K.,
Wyatt, N., Xiang, Y., Xie, R. C., Xue, Z., Yoshikawa, H., Zhang, J., Zhang, P., Zhao, Y., Zheng, L., Zheng, X.-Y., Zieringer, M., Zimmer,
L. A., Ziveri, P., Zunino, P., and Zurbrick, C.: The GEOTRACES Intermediate Data Product 2017, Chem. Geol., 493, 210-223,
https://doi.org/10.1016/j.chemgeo.2018.05.040, 2018.
Severmann, S., Johnson, C. M., Beard, B. L., German, C. R., Edmonds, H. N., Chiba, H., and Green, D. R. H.: The effect of plume
processes on the Fe isotope composition of hydrothermally derived Fe in the deep ocean as inferred from the Rainbow vent site, Mid-
Atlantic Ridge, 36°14′N, Earth and Planetary Science Letters, 225, 63-76, 10.1016/j.epsl.2004.06.001, 2004.
Tagliabue, A. and Resing, J.: Impact of hydrothermalism on the ocean iron cycle, Philosophical Transactions of the Royal Society A:
Mathematical, Physical and Engineering Sciences, 374, 20150291, doi:10.1098/rsta.2015.0291, 2016.
Tagliabue, A., Bowie, A. R., Boyd, P. W., Buck, K. N., Johnson, K. S., and Saito, M. A.: The integral role of iron in ocean
biogeochemistry, Nature, 543, 51-59, 10.1038/nature21058, 2017.
Tagliabue, A., *et al*: Hydrothermal contribution to the oceanic dissolved iron inventory, Nature, 3, 252-256, 10.1038/ngeo818
10.1038/NGEO818, 2010.
Toner, B. M., German, C. R., Dick, G. J., and Breier, J. A.: Deciphering the Complex Chemistry of Deep-Ocean Particles Using
Complementary Synchrotron X-ray Microscope and Microprobe Instruments, Accounts Chem. Res., 49, 128-137, 2016.
Toner, B. M., Fakra, S. C., Manganini, S. J., Santelli, C. M., Marcus, M. A., Moffett, J., Rouxel, O., German, C. R., and Edwards, K. J.:
Preservation of iron(II) by carbon-rich matrices in a hydrothermal plume, Nat. Geosci., 2, 197-201, 10.1038/ngeo433, 2009.
Trocine, R. P. and Trefry, J. H.: Distribution and chemistry of suspended particles from an active hydrothermal vent site on the Mid-
Atlantic ridge at 26-degrees-N, Earth and Planetary Science Letters, 88, 1-15, 10.1016/0012-821x(88)90041-6, 1988.
Ussher, S. J., Achterberg, E. P., Sarthou, G., Laan, P., de Baar, H. J. W., and Worsfold, P. J.: Distribution of size fractionated dissolved
iron in the Canary Basin, Marine Environmental Research, 70, 46-55, http://dx.doi.org/10.1016/j.marenvres.2010.03.001, 2010.
Vic, C., Gula, J., Roullet, G., and Pradillon, F.: Dispersion of deep-sea hydrothermal vent effluents and larvae by submesoscale and tidal
currents, Deep Sea Research Part I: Oceanographic Research Papers, 1-51, 2018.
Weber, T.: Southern Ocean Upwelling and the Marine Iron Cycle, Geophysical Research Letters, 47, e2020GL090737,
https://doi.org/10.1029/2020GL090737, 2020.
Wu, J. F., Wells, M. L., and Rember, R.: Dissolved iron anomaly in the deep tropical-subtropical Pacific: Evidence for long-range
transport of hydrothermal iron, Geochimica et Cosmochimica Acta, 75, 460-468, 10.1016/j.gca.2010.10.024, 2011.
Yucel, M., Gartman, A., Chan, C. S., and Luther, G. W.: Hydrothermal vents as a kinetically stable source of iron-sulphide-bearing
nanoparticles to the ocean, Nat. Geosci., 4, 367-371, 10.1038/ngeo1148, 2011.

