# Peer review of "Tracing differences in iron addition to the Mid-Atlantic Ridge valley between hydrothermal vent sites: implications for the addition of iron to the deep ocean"

_Biogeosciences, 2022_

## Author Response (AR1)

Response to reviews

**R1**
General comments

Lough et al. report the variability in dissolved iron to excess helium ratios (dFe:xs3He) across geochemically distinct hydrothermal vent sites along the Mid-Atlantic Ridge, and compare methods for estimating dFe:xs3He. This is important to investigate since dFe:xs3He derived hydrothermal Fe fluxes are used in global biogeochemical models. Hydrothermal vents were tracked during the expedition using temperature, salinity, light scattering, oxidation-reduction potential indicative of reduced chemical species, and dMn via flow injection. Total dissolvable and dissolved Fe/Mn and He were measured onshore.

The variability in estimated dFe:xs3He among methods was higher at stations close to the vent site, and the authors determined that using Mn:xs3He measured in the standard rosette cast (where He is measured) and using dMn to extrapolate xs3He in the trace metal cast is the most appropriate method for calculating dFe:xs3He ratios, since this can account for differences in CTD sampling position between the two separate casts. This methods comparison is valuable and will be useful for guiding future work. The authors furthermore document particulate-dissolved Fe exchange at most of their sites, suggesting hydrothermal dFe does not consistently behave conservatively, at least over short distances from the vent. Fe binding ligands are likely important in limiting the amount of dFe released from vents and stabilized in plumes.

The manuscript is very well written and arguments are clear. Below are minor comments.

Line 92 – Include specific dates of sampling for reference

Table 2 – If I am following correctly, the major difference between Method 1/2 and Method 3 is that He measurements were not obtained at the same sampling depths as trace metals, and interpolations are needed to be performed. In contrast, sampling depths in Method 1 are the same, though He and trace metals are measured from different casts. Given this, I am confused as to why the number of measurements collected from the standard rosette does not match up with the trace metal rosette in the second and fourth columns (e.g., n = 4:3 for Site 6) of Table 2. And why do the TMR sampling depths integrated in the 4th column not match the TMR sampling depths integrated in the 2nd column? Being as clear as possible in the caption will help readers less familiar with these calculations follow along.

Line 246 – What about TDMn? Any differences between TDMn and dMn, and evidence for Mn precipitation in the plume? I understand it should be minimal compared to Fe precipitation, but would be helpful to see it plotted in the supplemental section for reference

We thank reviewer one for their positive feedback on our manuscript.
We will include the dates of sampling on line 92 (22$^{nd}$ December 2017-27$^{th}$ January 2018) in the revised manuscript.
To clarify the contents of table 2, for all the of the sampling in this study trace metals and He are never collected at exactly at the same depth, as we relied on the real time signals from the sensors to decide when and where to take samples. The distance between standard and clean sampling depths is in the range of 2-30 m. You can see from figure 3 that the sampled depths do not match up exactly between the standard and clean casts and the sampling resolution is higher on the clean casts.
This contrasts with previous studies (Fitzsimmons et al., 2017; Resing et al., 2015; Saito et al., 2013) using He to assess inputs of trace metals from hydrothermal vents, where samples were taken at the same depth on the standard and trace metal clean sampling casts, then integrated within a specific depth range of the plume e.g. between 1000-2000 m. Even though

plumes sampled over a ridge will have shifted between casts as we demonstrate in figure 2. We argue that it's better to use the sensor data to determine sampling depths rather than using pre-determined evenly spaced sampling depths.

In table 2 the number of measurements used for integration from the standard rosette does not match up with the trace metal rosette in the second column because less He samples were taken than trace metal samples.  The number of trace metal measurements used between the second and forth columns do not match up exactly because there were more dMn measurements from the trace metal rosette than the standard rosette, hence because method 2 uses xsHe values derived from dMn measurements on the trace metals rosette, the number of integrated depths is higher for some rows.

TDMn samples were only collected at a couple of sites and were usually within or close to the analytical uncertainty of dMn. We will add TDMn values to Figure S3 and S5.

**R2**

Summary: This manuscript presents very important new results for dFe (soluble, colloidal), dMn, 3-He (and 3-He excess), and dissolvable particulate Fe from a diverse array of hydrothermal plumes along the North Mid-Atlantic Ridge. There is significant uncertainty regarding the relationships among these parameters at hydrothermal sources, especially the dFe/3He-excess ratio and the complex factors underpinning its systematics. These results promise critical insights that could contribute to better global estimates of hydrothermal-sourced Fe to the oceans, itself a major climatological unknown. The study uses two (or three) methods to derive and calculate Fe/He relationships and has the scope to contribute to several outstanding questions, especially the importance and scale of plume-sampling variability and whether the use of the differentially scavengable element Mn can improve difficult-to-measure Fe/He relationships.

The detailed review from reviewer 2 is much appreciated and we are grateful for their efforts in helping to improve this manuscript.

Several challenges prevent this manuscript from reaching its full potential. As written, the text is confusing and disjointed: an overuse of demonstrative pronouns and numerous incomplete sentences consistently interrupt the manuscript's logical flow. The methods section insufficiently describes the ratio-determining methodologies at the core of the findings. The abstract implies a major finding regarding the importance plume age but no such description is given in the text. The conclusions section includes multiple non-conclusion discussion points.

We are happy to make grammatical corrections and amendments to sentences in the revised manuscript.

Merging of the results and discussion sections has made the manuscript especially difficult to unpack. It has forced the authors to address important discussion points (comparing the methodologies to each other and their wider context) alongside more results-specific points (variability in the plumes-as-sampled, inter-cast differences, and results-specific caveats like rising plumes). Basin-scale implications and source-fluid specific discussion points (especially the latter, being sourced from previously published results) would be better-suited in a dedicated discussion section or two, i.e. after inter-site results and inter-method differences have been presented and resolved. The "Wider implications" section 4 attempts to serve as a discussion section in many ways, but at present does so inadequately.

We are also happy to make adjustments to the structure of the manuscript where it is appropriate to do so to improve the flow. Our rationale for combing the results and discussion was to keep the manuscript relatively short, so it remains more accessible to researchers planning future studies of hydrothermal systems. This is also the format of many recent ocean science articles published in biogeosciences.

I attempt to address the major points in the sections below. Technical/writing points are broken out separately at the end of the comment.
Abstract: Line 17: "plume age" is mentioned as the primary driver for the ranges in Fe/He ratios, but 'age' is not once mentioned in the manuscript body. If distance from vent source or some other combination of parameters is being used as a proxy for age, either implicitly or explicitly, these relationships must be clearly stated and explained.

Comment on Line 17: Throughout the manuscript we refer to distance which is frequently assumed to be a proxy for age *i.e.* the further away from the site of venting the older the age of the plume is (Fitzsimmons et al., 2017; Resing et al., 2015; Saito et al., 2013). As we show with our results from separate casts directly over the vent sites, distance from the seafloor is not a reliable proxy for age. However, on the wider sampling scales of 40 km it is safe to assume that stations further away from the vent sites are sampling older plume waters. We will clarify this throughout in the revised manuscript.

Methods: The "integration methods" and their relevant calculations and equations (especially those for Methods 1 and 2, but also perhaps 3) should be defined, presented, and described at some point in the Methods section proper. At least, a summary of their external sourcing and relevant assumptions should be given before the Results section begins. "Integration", I presume, is being done versus depth, but this point is not explicitly stated. Omission of these methods—or their mis-placement in other sections—makes it difficult for future users to derive their own results using Methods 1 and 2, and also to test and confirm the results being presented here.

Comment on calculations for Fe/xsHe: We do not feel it is necessary to include equations for integration as this is a common place calculation within this subject and a concept that is usually taught before higher education. We will amend the text to specify that integration is done versus depth. We separated the different ways of calculating Fe/xsHe from the methods section as we decided the methods section should be specific to sampling and analytical methods.

Paragraph at Line 185: "despite the ship maintaining" is repeated twice in this paragraph, and this redundancy highlights that the purpose and ordering of this section is a bit confused. Is this paragraph serving to discuss the differences between/among the sites or the differences in the two methods? It seems to be trying to both, so consider revising this paragraph (and the ones nearby) to have clearer intent.
Specifically, it would help to discuss the inter-site differences, inter-method differences, and inter-cast/sampling system differences independently and in turn. Related to the final one, it might be clearer throughout the manuscript to refer to differences "between casts" or "between casts at the same station" rather than between "sampling systems". Different casts that are separated by time (even those using the same sampling system) will face plume-position issues. The cast's timing, rather than the system itself, is thus the source of the uncertainty for these purposes. Inter-"system" differences would be more appropriate terminology when focusing on potential contamination- or blank-related results that differ

between sampling systems, i.e. TMR vs S(S)R. Many of these topics would be best presented in a dedicated results section (or even in the methods if they can be methodologically caveated).

We will delete the repeated phrase and revise the paragraph (starting at L185) to focus on differences in Fe/xsHe caused by the shifting of the plume relative to the position of the ship. We will change the text to discuss between casts rather than sampling systems and focus the text to clarify that the source of uncertainty comes from the timing of casts rather than the casts themselves as reviewer 2 rightly points out.

Section 3.2: The ordering of assessments introduced here (LSS/ORP, Mn/He, Fe/He) does not match the order they are then presented and discussed in the text that follows, unless the earlier mention of LSS/ORP in Figure 3 is being back-referred to. This introductory sentence would make more sense if moved earlier in the manuscript.

In section 3.2 the order of assessments refers to the order used to establish how successful the sampling cast has been in capturing the full extent of the hydrothermal plume over the vent site. Step 1, assessing the sensor profiles is done at sea at the time of sampling hence why it is referred to earlier. Whereas steps 2 and 3 can only be done once the sample analysis is complete, which is why they are introduced in section 3.2.

Section 4 intro paragraphs (4.0?): Despite the title of this section, the introductory paragraphs (lines 325 to 350) do not significantly discuss wider "implications" of the manuscript's actual findings, at least until the very end (beg. Line 344) and then only briefly. Most of this text serves more to introduce/revisit previously published field findings and different model designs being debated in the literature. While this text is not without merit, it might be helpful to number and rename this set of paragraphs as its own subsection ("4.1 Expectations based on prior results"), or at least move the major findings (beginning at Line 344) to the _beginning_ and then follow through with discussing their implications more directly, as the section title promises.

We will move the findings to the beginning of the paragraph for L325 to L350.

Section 4.1: I had difficulty following the logic of the first two sentences beginning at Line 365. TAG has the lower vent fluid Fe/H2S ratio—why should it precipitate more FeS2 if ratios greater than 1 are more likely to precipitate more FeS2 (as stated in the previous sentence)?

We apologise for this mistake on L365, it should read "Fe/H$_2$S <1" as FeS$_2$ nanoparticles will be more prevalent when sulphide is enriched relative to Fe.

The section is titled "What controls near-field dissolved Fe to Helium ratios [?]", but by the end of this section I did not feel the text had convincingly and logically addressed that question. Instead, the section presents a list of hypotheticals and loosely-connected points that dances around the findings.

We can change the title of section 4.1 to "what controls ridge axis dissolved Fe to helium ratios" to be more specific". The main two points of this section are:
1. When we look at our results in comparison to Fe$^{2+}$ oxidation (paragraph 1) and Fe/H$_2$S of the vents (paragraph 2) these parameters cannot fully explain differences in dFe/xsHe, we

therefore suggest these differences are primarily controlled by organic ligands (end of paragraph 2). As we have ruled out the possibility of differences in inorganic chemistry being the controlling factors.

2. The dFe/xsHe ratio measured within the ridge valley at the scale of 10-40 km is similar between sites to that used in global biogeochemical models. However, there is significant variability in the appPFe/dFe which has implications for the way hydrothermal Fe is modelled. Which leads into the next section.

We will revise section 4.1 for clarity however given reviewers one and three did not take issue with this section we think a complete re-write is unecessary.

Section 4.2: The major scientific conclusion of this section is that the fairly consistent 10–40 km-distance Fe/He relationships observed in Fig 5 are likely to be reflective of Fe/He ratios leaking out of the ridge into the deep N. Atlantic. State this from the very beginning, then justify it. The points summarizing these findings are not scientifically unsound as written, but they are surprisingly roundabout. The flow of the writing here is difficult to follow as the (very long) paragraph meanders amongst discussion of various assumptions, results/findings, hypotheticals, and conclusions.

The consistency of dFe/xsHe at the 10-40 km scale is stated at the end of section 4.1 which leads into 4.2. We will move it to the start of section 4.2 for clarity.

The topic sentence emblematically does not clearly state or frame any of the important points regarding Fe escape from the ridges that follow. Relatedly, the final, critical sentence in this section (and potentially the paper as a whole!)—that a significantly higher Fe/He ratio may be called for in global models, does not, at present, read as directly or well-supported by the lines preceding it.

We will edit the first paragraph of section 4.2 to emphasise that TDFe/xsHe was higher than dFe/xsHe and we therefore need to investigate the possibility that this difference will persist as plume waters are transported beyond the ridge. As this will impact the values used in global biogeochemical models.

Section 4.3: The concluding sentence here is a list of wide-ranging unknowns punctuated incorrectly by a question mark. It does not especially summarize or clarify any of the preceding points about future work.

The question mark at the end of section 4.3 will be removed. Two of the four points listed in this sentence (the frequency of vent systems and the variability in the hydrothermal ligand source) are the subject of the preceding paragraph.

Conclusions: Sentences from Line 448 to 452 are hypotheticals about future sampling systems—not conclusions. The final sentence of the paper is especially confusing—are the findings of this paper implying different or similar Fe limits at the various sites? Consider replacing most of this section with a clearer, more concrete summary of the actual conclusions reached as a result of this paper's new data, not hypotheticals based on what-ifs.

Sentences from Line 448 to 452 begin with conclusions based on our repeat sampling at the same vent site which highlights how variable depth profiles of a hydrothermal plume can be between casts taken hours apart. When it is usually assumed that one profile over a vent site is enough to constrain the concentration profile of a hydrothermal plume in the water column.

This is an important conclusion for the community and those that will be planning future sampling campaigns. We therefore feel it is prudent to recommend possible solutions to the difficulties of measuring Fe and He together in plumes so that other researchers can consider the technical issues when planning to study hydrothermal plumes using multiple sampling systems.

To clarify the final sentence of section 5, similarity in the near field dFe/xsHe relative to the vent Fe/xsHe shows there is a limit on the amount of Fe released from vents that can be converted into dissolved Fe in the water column. However, as a result of the scatter in the near-field dFe/xsHe we cannot say whether or not the this limit was higher or lower between the TAG and Rainbow vent sites, within the range of dFe/xsHe values measured (4-38 nmol/fmol).

We will revise section 5 in order to make the conclusions clearer.

Table 2: This table was a bit unintentionally confusing: only half the values tabulated are Fe/He ratios as labelled in the header. The other half are n values, which themselves are separated via colons (confusingly suggesting they are ratios). I suggest revisiting the table and column labelling to increase clarity. The organization of Table S1 is clearer in some respects as it groups like datatypes together (ratios, statistical parameters) allowing the two methods to be more easily compared.

We will swap table S1 for table 2 in the revised manuscript. Table S1 will now show the number of samples used for the integrations to keep this information separate and avoid confusion.

It is not possible to use Figure 1 to unambiguously determine which stations are which in Tables 2 and S1. In the case of the Lucky Strike site, for instance, only one station label is shown in Figure 1, but two rows (stations/casts?) are listed in Tables 2 and S1. Consider using some combination of station/cast numbers to unambiguously label Figure 1 and both tables. Relatedly, the descriptive labels in the lower half of Table S1 (e.g. "Close N of TAG") read as overly subjective, even for a supplement.

For figure 1 there are two points on the map at lucky strike, but they are so close together that they appear as one on the map. We will add station numbers in brackets to figure 1 and in table 2 and table S1 so they are more easily relatable. We will change the descriptions to include the distance e.g. change "Close N of TAG" to "29 km N of TAG (S26)"

The value of 35 nmol/fmol for Rainbow 38 (the instance with no asterisk) differs from the matching row reported in Table S1 (34 nmol/fmol). This is likely a significant figures issue? Given the importance of these data, I suggest revising both Tables 2 and S1 to ensure a consistent number of significant figures (three?) for the reported Fe/He ratio values.

We will revise the data in tables and report the data to 3 significant figures as requested.

Figure S1: The difference between dMn measurements ("I" = "in"?) for "surface" waters at station 25 appears at several points to be much greater than 0.2 nM, especially in the upper 100 m. I agree these differences are not especially significant at depths relevant to the manuscript, but more care needs to be taken here in describing this offset (inter-cast differences? Time of day of the casts?)

For figure S1, we will corret the typo ("I" = "in") and add further description to the caption that this offset could be the result of differences in the time of day that sampling took place.

Figure S3-S9: How was the "N. Atlantic background value" for dMn determined? It's fine if it's just a "typical background" value, but the rationale should be stated somewhere.

The N. Atlantic background value was determined from dMn measurements of waters at the same depth range to the samples collected in this study but from the GEOTRACES equatorial Atlantic (GA03) and western Atlantic (GA02) at open ocean stations away from any margin sources. Background dFe was determined in the same way. We will add text to explain this in the manuscript at L181 where the background values used are stated.

Are the very small numbers in the grey bars atop each subfigure cast numbers or station numbers? The captions seem to imply they are cast numbers ("Main casts are…"), but the main text (Table 2) refers to them as stations. Be consistent. Either way, the numbers listed in the captions and figures here are not enough to identify which site is which (e.g. 12–15; 26, 27?) without forcing the reader to cross-reference other tables/text. Consider grouping/labeling all these supplemental sub-figures by site.

The numbers in the grey boxes of supplementary figures are station numbers not cast numbers. We will correct this in the caption. We will change the labels to the new descriptions e.g. change "Close N of TAG" to "29 km N of TAG (S26)" that will be added to the tables. This should make comparison between these figures and the tables more straight forward.
* * *
Technical/Writing and Formatting Issues:

As described in the summary, the text as written suffers from an over-use of demonstrative pronouns ("this" "these"), especially at the beginning of sentences. I have tried to point out important instances in my line-specific comments, but the authors should revisit all uses of these terms and revise to use more specific phrasing whenever possible. Relatedly, the number of incomplete sentences and disconnected clauses is high, making it difficult to follow the logic at many points.
We will go through the manuscript and correct all of the points reviewer 2 has highlighted in their line by line comments and revisit all uses of the terms mentioned to see if more specific phrasing can be used to make it easier to follow the logic.
Line 99: Missing apostrophe in "plumes" (should be either plume's or plumes')
This will be corrected in the revised manuscript.
Line 127: Missing hyphen: "In-house"
This will be corrected in the revised manuscript.
Line 137: "Analyzed simultaneously during sample analysis" is redundant.
This will be corrected in the revised manuscript.
Line 152: Missing a hyphen: "near-impossible" (or "nearly impossible")
This will be corrected in the revised manuscript.
Line 157: "ratio's" should be "ratios"
This will be corrected in the revised manuscript.
Line 158: "off-axis" should be hyphenated
This will be corrected in the revised manuscript.

Line 174: For consistency, and even though it is not discussed at length, this third method should also probably be named (e.g. "3He-interpolation method"), rather than just "A third method…"

The naming will be changed in the revised manuscript.

Line 179: Consider changing "Thusly" to "We therefore"

This will be corrected in the revised manuscript.

Lines 185-186: The word "respectively" could be interpreted as referring to the either the two different integration methods or the two sites. Consider restructuring this sentence to reduce ambiguity, e.g. "…ranging from 4 to 87 at the TAG site and 4 to 63 nmol/fmol at Rainbow site."

We will restructure sentences to separate the TAG and Rainbow data as in the provided example and elsewhere in the manuscript where the term respectively has been used.

Lines 194/195: "down to" is an awkward construction "due to" it implying depth in the ocean. Use a different term.

This will be corrected in the revised manuscript.

Line 197: Incomplete sentence at "As it can account..."

This sentence will be revised.

Line 202 and 203: "over the vent sites" is somewhat unclear—do you mean "across different vent sites"? "Over the vent sites" sounds like a ship position/cast difference issue, as implied by the concluding words "…when comparing different casts". As pointed out in my comment re: paragraph at Line 185, it would be best to address methodological (Method 1 vs 2), inter-site (TAG vs Rainbow), and cast/sampling system differences (TAG TMR vs TAG SR) independently and systematically.

We will adopt the example naming strategy throughout to make it clearer when we are discussing differences between methods, sites and sampling casts.

Line 223: "over the vent sites" is again a confusing phrase here, as it implies the ship's position. Is might be clearer to say "The extent to which any collected sample is representative of the core…" (or similar)

We will add the provided example in this instance and look to use more precise language where the term over the vent site is used.

Line 240: Agreement issue: "Extrapolation…indicates"

This will be corrected in the revised manuscript.

Line 255: "its" should be "it is"

This will be corrected in the revised manuscript.

Lines 259-262: Comma usage in this sentence is confusing. Is "uncertainties associated with" relevant to both "vent fluid end-members" and the degree of removal of dMn at the two sites? In the following sentence, usage of "this" (and later "that") are unclear about what is being referenced: the sub-1:1-line relationships in general, or the specifics listed at the end of the previous sentence?

The mentioned uncertainties cover all of the possible reasons why points on the graph fall above or below the 1:1 line. We will edit the sentence to be more direct removing usage of "this" and "that".

Line 268: "and is" (singular) doesn't agree with the first half of the sentence.

"and is" will be replaced with which and the sentence revised.

Line 272: Agreement issue: "site-by-site differences…was not simply"

This will be revised

Line 283: Consider being more specific about these spatial scales right off the bat: "Over short spatial scales of under 40 km from the vent site, …" The sentence that follows is largely redundant.

We will edit and combine these two sentences to be more succinct as suggested

Line 286: Hyphenate "vent-derived"

This will be corrected in the revised manuscript.

Line 290: To what does "This [indicates]" refer? The outlier discussed in the previous sentence?

The outlier in the previous section is in brackets. The "this" refers to the drop in dFe/xsHe between the vent stations at 0 km and the stations in the 10-40 km range which is the main subject of the sentence.

Line 293-295: As written, this sentence wrongly implies that the Fe/He ratios are "observing" the wide range of TDFe concentrations, rather than the authors.

We will reword the sentence and remove the term observing.

Line 298-300: "West" is unnecessarily capitalized. Without additional information, it is not clear what "log K" specifically refers to here. Again, the usage of "This [could explain]…" is confusing and needs to be clarified.

Log K refers to the oxidation rate constant of Fe(II). We will revise the sentence to "had anomalous Fe(II) oxidation rate constant values (log K),"

Line 309: What does 'This [highlights]…" refer to? The cFe/dFe ratios, or the appPFe/dFe ratios? (Or both?) Revise to be clear.

The difference in the appPFe/dFe ratio, we will revise to make this clearer.

Line 316: The sentence beginning "Specifically…" is incomplete.

This sentence will be revised for clarity

Lines 321-322: Two sentences in a row begin with "this" here, making the logic difficult to follow.

Second this will be changed to which.

Line 326: "short" spatial scales is rather non-specific. Consider replacing or supplementing with the actual spatial scales (0– or 10–40 km, I presume). Also, the phrase "short spatial scales between the TAG and Rainbow plumes" implies the distance between the TAG and Rainbow sites rather than the distances over which each site's plume was sampled.

We will go through the manuscript and replace the terms short, distal and near field with specific distances.

Line 336 (and 355): Hyphenated "Fe-binding ligand[s]"

Line 345: Hyphenated "basin-scale"

Line 349: Hyphenated "particulate-dissolved Fe exchange" and "smaller-scale"

Line 355: "strength to" is an incomplete thought/clause removed "to"

Line 365: corrected to "molar ratios of Fe/H2S >1"

Line 372: The sentence beginning "Suggesting…" is incomplete.
This sentence will be revised for clarity

Line 385: The sentence beginning "Showing…" is incomplete.

Line 387: Hyphenated "Fe-rich". Explicitly state or otherwise clarify the "residence time" being referred to from the Vic reference. The residence time refers to the time it takes for tracer lagrangian particles within a physical mesoscale model to exit the ridge valley. This will be clarified in the text.

Lines 389-395: How many times was Stokes' law actually used by the authors here? Twice it is mentioned, but only one calculation is discussed (though not explicitly shown), which reads as redundant.
It was only used once to assess the potential for FeOOH particles to settle out of the plume during further dispersion beyond the sampled 0-40 km range. We will edit the text to make this clear.

Line 406: "This [would counteract]…" is unclear: which of the previously listed items (or all?) is being referred to? It refers to both, will change "this" to "these forces"

Line 410: "This [creates]…" is unclear. This sentence will be deleted

Line 414: Multiple things are listed and then referred to as "…is key to determining"
"are key to determining"

Line 415: "This [highlights]…" is, again, awkward and difficult to follow. Revised to "It follows"

Line 428: "maybe" should be "may be". Corrected to may be

Figure 1: The text labels on many of the sub-figures are on the smaller side. Font size may need to be increased for publication.
This formatting follows the biogeosciences journal word document template
Figure 2, 4, and 5 captions: It would be clearer if the figure sub-identifiers ("(a)", "(b)"…) were listed before (rather than after) their respective text descriptions. Currently, succinct descriptors and ancillary information are mixed throughout the caption text, making these captions hard to follow.
We will revise the caption and move letters before the description.
Table S1: "Plume-integrated" should probably be hyphenated. The word "values" in the header is extraneous. The vent sites in this table (cf. comments re: Table 2 in the main text) lack station numbers and cannot be identified in the Figures or directly compared to other tables.
See previous comments on revising this table, the term values will be removed
The text of the first paragraph of Table S1's description confusingly uses "this" several times, referring to different things each time. Please revise to be more precise. The text of the second paragraph also needs revision. Specifically, "The estimated ratios from interpolation are 'so variable'…" reads as overly subjective, and the next sentence ("Largely because of the variability…") is long and awkwardly constructed.
This section will be revised to for clarity
Figure S2 caption: "Demonstrating…" is not a complete sentence. Merge this fragment with the topic sentence here—it is the major take-away.
This will be revised as recommended.
Figures S3-S9: Much of the in-figure text, with the exception of the y-axis labels, is very small. The font sizes may need to be enlarged for publication.

The font sizes of figures follow the biogeosciences guidelines.
Figure S5 caption: This caption mistakenly refers to Figure 3A, but the relationships discussed are shown in Figure 4A.
This mistake will be corrected
Figure S10: There is a missing close parenthesis in the first sentence of the caption. Spell out "three" or use "n=3" in this sentence as well. Remove the redundant "over Rainbow" from the second sentence.
This will be revised as recommended.
Figure S11: Font size in the grey boxes is much too small to be legible. Consider changing "which is what the main text focuses on" to something less chatty, e.g. "as are the focus of the main text".
The font will be increased to make the numbers clear and caption edited as recommended.

Author contributions: The lead author is referred to as both AL and AJML
It should be AJML, this will be corrected in the acknowledgements.

**R3**
The manuscript entitled " The impact of hydrothermal vent geochemistry on the addition of iron to the deep ocean." by Lough et al., present an important issue regarding supply of iron into world oceans through hydrothermal sources. Authors have tried to present the impacts of vent geochemistry on such contribution. For that, authors have presented a few vent-specific case studies about the behaviours of hydrothermal iron in deep oceanic environments along the MAR. They have estimated the excess dissolved iron to dissolved helium-3 ratios and exchanges of iron between dissolved & particulate fractions around those vent fields and which would improve our understanding about dispersion/contribution of iron from active hydrothermal sources into deep oceanic waters. The observed variability of those parameters have also nicely correlated with the other geochemical factors of particular vent fluid. It's also nice to see that authors have mentioned critically about the possible error sources for estimation (and therefore misinterpretation) of ex-Fe/3He ratios in deep water columns. This concept would be useful for similar types of observation in future. According to me, in the present manuscript, except few technical issues, the overall presentation of the topic is fine.
We thank reviewer three for their helpful comments.
The language used in many parts of the text are not very clear, and need some changes to make it more understandable to readers. The longer statements should split into simple shorter ones. Some of the other major lacking's are as follows:
We will revise the manuscript and split sentences into shorter ones where possible to improve clarity.
The topic discussed in this manuscript is based on observations at the only four vents in norther MAR. However, the title of manuscript was made in very generalise way, as if the draft has information's of vents from global oceans. Author may think about this and may slightly modify the title accordingly.
We will adjust the title to: The impact of hydrothermal vent geochemistry on the addition of iron to the deep ocean: case study of the northern Mid-Atlantic Ridge.
In abstract there is mentioned …" two methods of estimation…." for checking variability in dFe/xs3He ratios. Is this really mean two separate methodologies?? Or indicates two types of hydrocasts which were made during sampling?? If these two different types of hydrocasts, author should mention the details of operation and significance of those casts in the "sample collection" section (i.e., section 2.1).
The two methods of estimation refer to the two methodologies of used to calculate Fe/xsHe ratios. A third methodology is assessed briefly but as this method produced negative numbers

the results are in the supplement, and it is not discussed at length. The two different types of sampling equipment used are described on line 102 and 116. We will go through the manuscript and make sure the terminology is consistent throughout to make sure this is clear.
In methodology, authors should mention about the techniques (with necessary references) used for helium isotope analyses in water samples at WOI. Those helium analyses details should appear in "Sample analysis" section (section 2.2.); instead of section 2.1. The

We will add a brief description of the helium isotope analysis to section 2.2

Even the vent sites are very well known, still it is better to mention about the sampling station locations (lat, long); may be in Table 1.

We will add sample station number to figure 1 and table 1 which was also requested by reviewer 2.

Initially in the abstract and introduction authors are mentioned studies are carried out four vent field with different geochemistry. But latter it has found that the results and discussion (and figure) are restricted to the N-MORB hosted Rainbow and TAG fields only and the E-MORB hosted Lucky Strike and Menez Gwen fields are excluded- any specific reason?? In result and discussion section, the water column profiles of xsHe, dFe, dMn from the Rainbow are presented. What about the profiles of other three fields? For geochemical comparison those data of other fields are essential.

Much of the manuscript focuses on TAG and Rainbow as a wider range of distances were sampled from around those vent sites, so we could compare the separation of iron into colloids and particles between those two sites over the same distance. For Menez Gwen and Lucky Strike we only sampled directly over the vent site. As a result, the discussion of results from lucky strike and Menez Gwen is restricted to section 3.2.

In figure 3 we present the depth profiles of elements and sensors in order to highlight the real consequences of the concept shown in figure 2. Which is that the plume shifts between sampling casts and the full extent of the plume over a vent site cannot be captured without sampling based on sensor signals and ideally multiple sampling casts.

The depth profiles from all stations are presented in the supplement to avoid overloading the manuscript with multi panel plots.

It looks the dFe/H2S ratios in the text and table might have an unit of nmol/mmol.

The units for dFe/H2S are mmol/mmol, this will be added to the table 1 to make it clear.

The first paragraph of conclusion (Ln: 447-455) which mostly mentioned about technical suggestion for deep water sampling doesn't looks very essential remarks to address about any "impacts of hydrothermal geochemistry on supply of iron" in deep ocean waters.

Whilst the issue of obtaining representative samples of hydrothermal plumes is not mentioned in the title of the paper we do feel it is a key and often overlooked subject within this area of research. We therefore think it is necessary to highlight the need for hydrothermal sampling programs led by sensor data, as well as providing technical solutions so that future research programs will be able to trace hydrothermal inputs from the vent source beyond the near field and into the open ocean.

Fitzsimmons, J.N., John, S.G., Marsay, C.M., Hoffman, C.L., Nicholas, Sarah L., Toner, B.M., German, C.R., Sherrell, R.M., 2017. Iron persistence in a distal hydrothermal plume supported by dissolved–particulate exchange. Nature Geoscience 10, 195.

Resing, J.A., Sedwick, P.N., German, C.R., Jenkins, W.J., Moffett, J.W., Sohst, B.M., Tagliabue, A., 2015. Basin-scale transport of hydrothermal dissolved metals across the South Pacific Ocean. Nature 523, 200-U140.

Saito, M.A., Noble, A.E., Tagliabue, A., Goepfert, T.J., Lamborg, C.H., Jenkins, W.J., 2013. Slow-spreading submarine ridges in the South Atlantic as a significant oceanic iron source. Nature Geoscience 6, 775-779.

---

## Author Response (AR2)

Response to reviews 2

We would like to thank all the reviewers for their efforts in improving the manuscript for publication.

We have corrected the minor typing errors identified by reviewer 1, we have also thoroughly checked the manuscript for additional typing errors and grammar. We have amended section L430-435 as we identified an error in the particle settling calculation that has now been corrected. The identified error does not substantially change the content of the manuscript or interpretation of the results. A summary table of the iron particle diameters gathered form the literature has also been added to the supplementary information.